# Modeling Unseen Environments with Language-guided Composable Causal Components in Reinforcement Learning

**Xinyue Wang[1], Biwei Huang[1]**
[1]University of California San Diego
{xiw159, bih007}@ucsd.edu

## Abstract

Generalization in reinforcement learning (RL) remains a significant challenge, especially when agents encounter novel environments with unseen dynamics. Drawing inspiration from human compositional reasoning—where known components are reconfigured to handle new situations—we introduce World Modeling with Compositional Causal Components (WM3C). This novel framework enhances RL generalization by learning and leveraging compositional causal components. Unlike previous approaches focusing on invariant representation learning or meta-learning, WM3C identifies and utilizes causal dynamics among composable elements, facilitating robust adaptation to new tasks. Our approach integrates language as a compositional modality to decompose the latent space into meaningful components and provides theoretical guarantees for their unique identification under mild assumptions. Our practical implementation uses a masked autoencoder with mutual information constraints and adaptive sparsity regularization to capture high-level semantic information and effectively disentangle transition dynamics. Experiments on numerical simulations and real-world robotic manipulation tasks demonstrate that WM3C significantly outperforms existing methods in identifying latent processes, improving policy learning, and generalizing to unseen tasks.[1]

## 1 Introduction

Reinforcement learning (RL) has rapidly progressed, driving innovations in domains such as game playing, robotics, and autonomous driving (Silver et al., 2018; Vinyals et al., 2019; Shi et al., 2022; Kiran et al., 2020). Deep reinforcement learning (DRL) methods, including Deep Q-Networks (DQN), Soft Actor-Critic (SAC), and Proximal Policy Optimization (PPO), have addressed various challenges in RL, such as stability in training, exploration in large state spaces, and efficient policy optimization (Haarnoja et al., 2018; Schulman et al., 2017; Mnih et al., 2015; 2016; Fujimoto et al., 2018). These breakthroughs underscore the pivotal role of DRL in advancing artificial intelligence.

Despite these substantial advancements, one of the most pressing issues of DRL is the generalization of learned policies to novel, unseen environments (Gamrian & Goldberg, 2018; Song et al., 2019; Cobbe et al., 2018). For example, the policy excels in *push ball to place A* might perform notoriously poorly in the task *push ball to place B*. This limitation is primarily due to overfitting to specific training environments. Especially when the agent can only receive visual input in a partially observable environment, capturing the changes of the observation function and reward function in the new environment becomes even harder. Thus, locating the change in unseen environment and adapting learned knowledge to accommodate it are crucial for a generalizable agent.

Previous methods address the challenge of accommodating changes from different perspectives. Methods like data augmentation and visual encoders improve model robustness to observation function changes by attempting to incorporate potential visual changes in the training domain (Lee et al., 2019; Hansen & Wang, 2020; Yuan et al., 2022; Nair et al., 2022). However, these methods of-

---

[1]Project website: https://www.charonwangg.com/project/wm3c

ten rely on extensive domain knowledge to design effective augmentations and can struggle with changes that were not anticipated during training, e.g. changes in the state space and its dynamic structure. Additionally, approaches such as invariant representation learning and meta-reinforcement learning learn task-agnostic information that is stable across multiple training tasks to generalize to novel observations (Zhang et al., 2020b;a; Duan et al., 2016; Finn et al., 2017). While the former extracts invariant features that improve the agent's performance in new environments by leveraging consistent and transferable information, it may be limited in its ability to handle environments with fundamentally different underlying dynamics. The latter optimizes meta-parameters that enable rapid adaptation and efficient learning in novel tasks, but it can be computationally expensive and may not always guarantee quick convergence in highly variable environments. Meanwhile, advances in model-based reinforcement learning, including DreamerV3 (Hafner et al., 2023) and TD-MPC2 (Hansen et al., 2023), exhibit hyper-parameter robustness abilities across different kinds of environments. Despite their data efficiency, the learned world model may not accurately capture all aspects of the underlying data generation process, leading to suboptimal planning and poor generalization to new situations.

To this end, we ask the question: *What does it take to learn a generalizable world model?* Humans understand new concepts and generalize from known structures to unknown situations in a compositional way (Tenenbaum et al., 2011; Marcus, 2003; Gentner & Markman, 1997). For example, we learn *push ball to place A* as learning the composable components *push*, *ball*, *place A* and their relationships rather than as a single, indivisible task. This compositional understanding allows us to apply the learned components to new contexts, such as *push puck to place A*, reusing familiar elements in a new configuration. By learning these components and their dynamics, we can efficiently adapt to new tasks and environments.

A critical aspect of achieving this compositional generalization is understanding the environment's causal structure. A causal system is characterized by modularity and sparsity (Pearl, 2009; Peters et al., 2017; Glymour et al., 2019), which is naturally compositionally generalizable. This means that each component can be learned independently, and then recombined with minimal changes to handle new tasks. For instance, understanding the causal relationship between the action *push* and the object *ball* in one scenario can be reused to understand the relationship between 'push' and 'puck' in another. This modular approach aligns with human learning, where we reuse learned components across different contexts, enhancing our ability to generalize.

By drawing the connection between the causal system and compositional generalization, we propose World Modeling with Compositional Causal Components (WM3C), a framework that enables the agent to identify composable components, learn the causal dynamics among them, and utilize them for efficient training and adaptation. While previous work has explored causal representation learning to enhance reinforcement learning (Huang et al., 2021; Liu et al., 2023; Feng & Magliacane, 2024), none have focused on improving generalization via learning composable causal components and understanding their dynamics. To identify these composable causal components, we leverage another compositional modality, language (Fodor & Pylyshyn, 1988). We theoretically show that under mild and reasonable assumptions, composable causal components and their dynamics can be uniquely identified. We further provide an algorithm for learning a world model that incorporates these components and demonstrate how this approach generalizes to unseen tasks. The effectiveness of our approach is validated by achieving state-of-the-art performance on a numerical simulation dataset and a collection of robot manipulation in the Meta-World environment.

## 2 WORLD MODEL WITH COMPOSITIONAL CAUSAL COMPONENTS

In the following, we start by giving the motivation and intuition behind our formulation. Then, we present the identification theory that guarantees their correct identification. We further demonstrate the world model learning framework based on the environment model and our identification results.

### 2.1 MOTIVATION AND INTUITIONS

Consider a transfer learning scenario in visual-based RL, where an agent interacts with various familiar environments and then adapts to new, unseen ones. The challenge here is not just mastering individual tasks but leveraging past experiences to quickly adapt to novel tasks. Humans excel at this

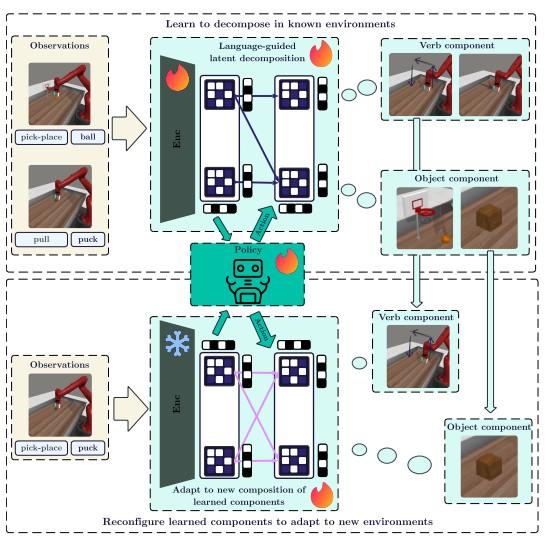

(a) Composable view of environment models.

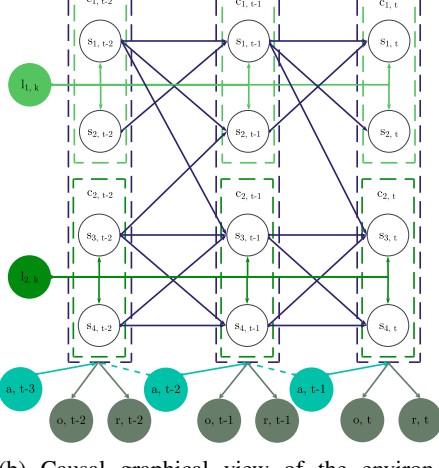

(b) Causal graphical view of the environment model. While the number of language-controlled components can be arbitrary, we present the framework of two components without loss of generality.

Figure 1: Illustrations of environment models

by abstracting composable concepts from known tasks and reusing them in new contexts through compositional generalization. In this process, language often plays a key role, naturally encapsulating the causal relationships and compositional structures of the environment (Fodor & Pylyshyn, 1988). For example, from tasks like *pick-place ball* and *pull puck*, we abstract the concepts of *pick-place* and *puck*. When faced with the task *pick-place puck*, we only need a few samples to combine these learned components—such as adjusting our focus and gesture to *pick up the puck* and place it in the desired location. This can be seen as a system with two latent components: verb (*pick-place*) and object (*puck*), whose values can be flexibly recombined in new domains for efficient adaptation.

This process mirrors the core principles of a causal system: sparsity and modularity (Pearl, 2009; Peters et al., 2017; Glymour et al., 2019; Schölkopf et al., 2021). Modularity enables us to independently identify causal components and recombine them in novel ways, facilitating adaptation to different systems. Sparsity ensures that when a distribution shift occurs, it minimally alters the system, leaving most components unchanged. Thus, only a few new samples are needed to adapt to new compositions of familiar components. For instance, to generalize efficiently to *pick-place puck*, we only need to adjust the causal dynamics between *pick-place* and *puck*, while the rest of the system remains largely intact.

Motivated by modeling the composable environment model from a causal view, we characterize the environment model using an augmented graph of the partially observable Markov decision process (See Figure 1b). We denote the sequence of observations as $\{< o_t, a_t, r_t >\}_{t=1}^{T}$, referring to image, action, and reward. We denote the underlying latents by $\boldsymbol{s}_t = (s_{1,t}, s_{2,t} \ldots s_{d,t})$ and $N$ language components as $\{l_1, \ldots, l_m\}$. We further assume that $\boldsymbol{s}_t = (\boldsymbol{c}_{1,t}, \ldots, \boldsymbol{c}_{m,t})$ can be uniquely partitioned into $m$ disjoint language-controlled components. The dimension of the language-controlled components $\boldsymbol{c}_i$ is denoted as $n_{\boldsymbol{c}_i}$. The following proposition establishes how we identify these components:

**Proposition 1** (Language-Controlled Components). *Under the assumption that the graphical representation of the environment model is Markov and faithful (Spirtes et al., 2001; Glymour et al., 2019; Pearl, 2009) to the data, $\boldsymbol{c}_{i,t}$[2] is a minimal subset of state dimensions that are directly controlled by the language component $l_i$ and $s_{j,t} \in \boldsymbol{c}_{i,t}$ if and only if $s_{j,t} \not\!\perp\!\!\!\perp l_i \mid a_{t-1:t}, \boldsymbol{s}_{t-1}$, and $s_{j,t} \perp\!\!\!\perp \{l_k\}_{k \neq i} \mid l_i, a_{t-1:t}, \boldsymbol{s}_{t-1}$.*

---

[2]$\boldsymbol{c}_{i,t}$ and $\boldsymbol{c}_i$ are used interchangeably in this paper.

Intuitively speaking, language-controlled components are a set of state dimensions that are directly controlled by the individual language components, and this allows us to identify a modular and interpretable environment model. We can achieve compositional generalization in the latent space with the guidance of language components, in the unseen test domains. We further relax the conditions to allow the individual component to be influenced by all previous states instead of only the previous itself (see Equation 2). The data generation process based on composable components is mathematically formulated as follows:

$$[o_t, r_t] = g(\boldsymbol{s}_t, \epsilon_t), \quad \boldsymbol{s}_t = (\boldsymbol{c}_{i,t}, \ldots, \boldsymbol{c}_{m,t}), \quad L = \{l_1, ..., l_m\} \tag{1}$$

$$\boldsymbol{c}_{i,t} \sim p(\boldsymbol{c}_{i,t} | l_i, \boldsymbol{s}_{t-1}, a_{t-1}) \quad \text{for } i = 1, \ldots, m \tag{2}$$

## 2.2 IDENTIFIABILITY THEORY

Accurately identifying composable components and causal dynamics is crucial to developing robust models capable of generalizing across diverse environments. However, this task is challenging due to the inherent uncertainty and complexity of the data generation process. Previous work in non-linear ICA (Khemakhem et al., 2019; Hoyer et al., 2008; Hyvärinen & Pajunen, 1999; Klindt et al., 2020; Hyvarinen & Morioka, 2016) develops methods for identifying latent variables dimension-wise using strong assumptions like independence of noise terms and specific functional form priors. However, they often struggle with complex systems having interdependent latent structures and are limited by their reliance on a single auxiliary variable, parametric assumptions, and simple graphical models. Additionally, their sophisticated optimization procedures are difficult to scale up in real-world applications.

We focus on block-wise identifiability instead of dimension-wise identifiability to achieve a better trade-off between scalability and estimation accuracy. Our approach allows separate language components to independently control their corresponding latent variables, unlike previous temporal methods (Yao et al., 2021; 2022; Song et al., 2024) that rely on a single auxiliary variable connecting with all latent variables. By focusing on identifying language-controlled composable components without parametric assumptions, we can maintain theoretical guarantees while requiring significantly fewer language component values for identification. Note that this framework extends beyond language as a compositional modality - it applies to any latent system with multiple intermittent control signals, such as decomposing robotics tasks into components controlled by various signals. We believe this is the first work to demonstrate the identifiability of disentangled components separately controlled by different intermittent control signals in a general non-linear case for reinforcement learning tasks.

We define block-wise identifiability concerning the identification of the language-controlled component as follows. Some proof techniques and notations are related to Liu et al. (2023); Sun et al. (2024).

**Definition 1** (Block-wise Identifiability) *The true components of changing variables $\boldsymbol{c}_i$ are block-wise identifiable if, for the estimated component of changing variables $\hat{\boldsymbol{c}}_i$ and each component of changing variables $\boldsymbol{c}_i$, there exists an invertible function $h_i : \mathbb{R}^{n_{\boldsymbol{c}_i}} \to \mathbb{R}^{n_{\boldsymbol{c}_i}}$ such that $\boldsymbol{c}_i = h_i(\hat{\boldsymbol{c}}_i)$.*

**Theorem 1** *Suppose that the data generation process follows Equation 1, 2, and the following assumptions are fulfilled, then the language-controlled component $\boldsymbol{c}_i$ is block-wise identifiable:*

1. *The mixing function $[o_t, r_t] = g(\boldsymbol{s}_t, \epsilon_t)$ is invertible and smooth.*

2. *The set $\{s_i \in S \mid p(s_i) = 0\}$ has measure zero.*

3. *The conditional probability density should be sufficiently smooth, i.e., $p(s_{i,t} | l_i, \boldsymbol{s}_{t-1}, a_{t-1})$ is at least first-order differentiable.*

4. *Given language components $l_i$, previous state $\boldsymbol{s}_{t-1}$ and previous action $a_{t-1}$, every element of latent variable $s_{j,t}$ should be independent of each other, i.e., $s_{j,t} \perp\!\!\!\perp s_{k,t} \mid l_i, \boldsymbol{s}_{t-1}, a_{t-1}$ for $j, k \in \{1, \ldots, n\}$ and $j \neq k$.*

5. *For any $\boldsymbol{c}_i \in C_i$, we assume that there exist $n_{\boldsymbol{c}_i} + 1$ values of $l_i$ such that for $j = 1, \ldots, n_{\boldsymbol{c}_i}$ and $k = 1, \ldots, n_{\boldsymbol{c}_i}$, the following matrix is invertible:*

$$\begin{bmatrix} \varphi_1'(1,0) & \cdots & \varphi_j'(1,0) & \cdots & \varphi_{n_{c_t^i}}'(1,0) \\ \vdots & \ddots & \vdots & \ddots & \vdots \\ \varphi_1'(k,0) & \cdots & \varphi_j'(k,0) & \cdots & \varphi_{n_{c_t^i}}'(k,0) \\ \vdots & \ddots & \vdots & \ddots & \vdots \\ \varphi_1'(n_{c_t^i},0) & \cdots & \varphi_j'(n_{c_t^i},0) & \cdots & \varphi_{n_{c_t^i}}'(n_{c_t^i},0) \end{bmatrix}$$

*where $\varphi_j'(k,0) := \frac{\partial \log p(s_{j,t}|l_{i,k},\boldsymbol{s}_{t-1},a_{t-1})}{\partial s_{j,t}} - \frac{\partial \log p(s_{j,t}|l_{i,0},\boldsymbol{s}_{t-1},a_{t-1})}{\partial s_{j,t}}$ is the difference of the first-order derivative of the log density of $s_{j,t}$ between the $k$th value and $0$th value of the language component $l_i$.*

Most of these assumptions are commonly made in the field of causal representation learning (Kong et al., 2023a; Von Kügelgen et al., 2021; Huang et al., 2022; Liu et al., 2023; Sun et al., 2024). They help prevent degenerate cases, ensuring that the composable components in the model are in generic conditions, while enabling causal structure recovery - once latent variables are identified, their causal relationships can be discovered with standard causal discovery methods as used in Yao et al. (2021; 2022). The above theorem introduces a relaxed form of identifiability, showing that for each language-controlled composable component $\boldsymbol{c}_i$, with $n_{\boldsymbol{c}_i} + 1$ distinct values in a given language component (e.g., $n_{\boldsymbol{c}_i} + 1$ objects in *object* component), each true changing variable can be represented as a function of all estimated changing variables. It suggests that the estimated changing variables encapsulate all the necessary information for the true changing variables, effectively separating the component corresponding to the language component $\mathcal{L}_i$ and the other components not controlled by it. The minimum number of tasks for identifying all $m$ language-controlled components can be as low as $\sum_i^m n_{\boldsymbol{c}_i} + 1$. More discussions about the requirements of identifying all language-controlled components $\{\boldsymbol{c}_1, ..., \boldsymbol{c}_m\}$ and detailed proof are in Appendix A.3.

## 2.3 LEARNING COMPOSABLE WORLD MODELS

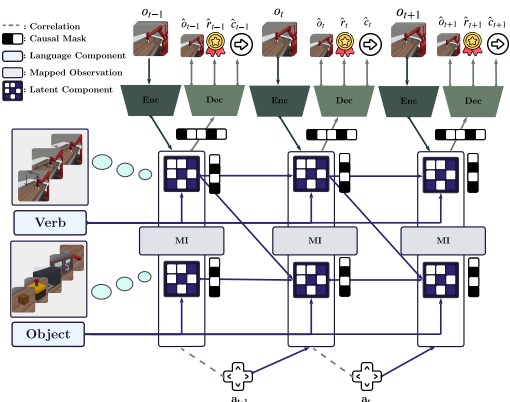
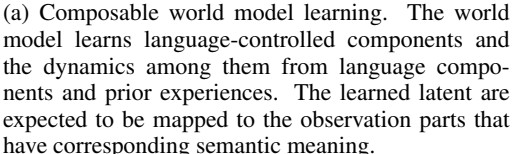
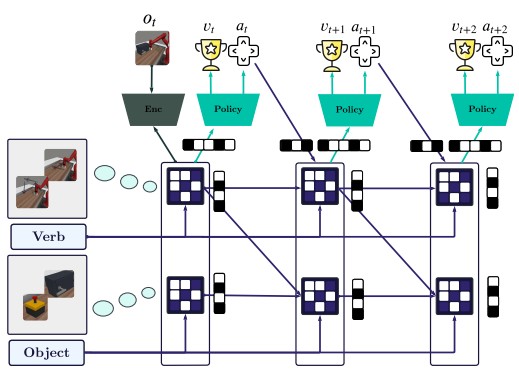

(a) Composable world model learning. The world model learns language-controlled components and the dynamics among them from language components and prior experiences. The learned latent are expected to be mapped to the observation parts that have corresponding semantic meaning.

(b) World model imagination and policy learning. Prompting by the language component values and the intial observations, the world model composes imaginations with language-controlled components and only the reward-related states are used for policy learning.

Figure 2: Illustrations of environment models.

Our world model identifies composable components and learns composition rules through agent interactions. The learning procedure is grounded on the identifiability results and incorporates natural properties of causal systems to improve learning and generalization. Notably, our learning framework is agnostic. To create a more efficient prototype, we utilize the state-of-the-art model-based reinforcement learning algorithm, DreamerV3 (Hafner et al., 2023), and apply insights from the identifiability results to guide the learning process.

**Approximating invertible mixing function.** The traditional encoder-decoder architecture is adopted to fulfill the invertibility of the mixing function given in Assumption 1. We also learn to predict the continuation signal for policy learning. To enhance practicality in the multi-task setting, we use a learnable task embedding $z$ to encode each task $u$, helping these decoding functions to capture variations across domains (see Figure 2a). We present two variants of WM3C, one uses a convolutional neural network (CNN) as the default observation encoder and decoder, and the other employs a Masked Auto-encoder (MAE) following Seo et al. (2023). MAE is particularly effective at extracting high-level semantic information, which may help bridge the gap between language components and their corresponding latent components (Kong et al., 2023b). The model components are:

$$
\begin{cases}
\text{Task Encoder:} & z = f_\alpha(u) \\
\text{Observation Encoder:} & h_t \sim p_\beta(h_t \mid o_t) \\
\text{Observation Decoder:} & \hat{o}_t \sim p_\theta(\hat{o}_t \mid m_o \odot \boldsymbol{s}_t, z) \\
\text{Reward Decoder:} & \hat{r}_t \sim p_\theta(\hat{r}_t \mid m_r \odot \boldsymbol{s}_t, z) \\
\text{Continuation Decoder:} & \hat{c}_t \sim p_\theta(\hat{c}_t \mid m_c \odot \boldsymbol{s}_t, z)
\end{cases}
\tag{3}
$$

and the corresponding loss function is:

$$
l_{\text{rep}} = \mathbb{E}\left[\sum_{t=1}^{T} -\log p(o_t \mid m_o \odot \boldsymbol{s}_t, z) - \log p(r_t \mid m_r \odot \boldsymbol{s}_t, z) - \log p(c_t \mid m_c \odot \boldsymbol{s}_t, z)\right].
$$

**Facilitating modular dynamics.** We decompose both the transition model and the representation model into disentangled modules, following the causal structure of the environment model. The KL regularization is factorized into component-wise KL divergence as well, acting as a soft independence constraint that encourages the disentanglement of composable components. Moreover, the language components $l_i$ are encoded as token embeddings $e_i$ to increase model flexibility.

$$
\begin{cases}
\text{Language Component Encoder:} & e_i = f_\alpha(l_i) \\
\text{Component } \boldsymbol{c}_{i,t} \text{ Representation Model:} & \boldsymbol{c}_{i,t} \sim q_\gamma(\boldsymbol{c}_{i,t} \mid h_t, e_i, \boldsymbol{s}_{t-1}, a_{t-1}) \\
\text{Component } \boldsymbol{c}_{i,t} \text{ Transition Model:} & \hat{\boldsymbol{c}_{i,t}} \sim p_\phi(\hat{\boldsymbol{c}}_{i,t} \mid e_i, \boldsymbol{s}_{t-1}, a_{t-1})
\end{cases}
\tag{4}
$$

$$
l_{\text{trans}} = \mathbb{E}\left[\sum_{t=2}^{T}\sum_{i=1}^{N} \text{KL}\left(q_\gamma(\boldsymbol{c}_{i,t} \mid h_t, e_i, \boldsymbol{s}_{t-1}, a_{t-1}) \;\|\; p_\phi(\boldsymbol{c}_{i,t} \mid e_i, \boldsymbol{s}_{t-1}, a_{t-1})\right)\right].
$$

**Enhancing causal structure.** The identification results imply that if we have enough values for each component, we can identify the component without additional constraints. However, we observe that it can be challenging in practice when the effect of the language signal is small. To address this, we add a mutual information (MI) constraint to strengthen the conditional independence described in the environment model (Belghazi et al., 2018). Specifically, for each language-controlled composable component, a mutual information neural estimator is used to maximize the joint MI between the component and its corresponding language component, while $n-1$ estimators minimize the joint MI between the component and other language components; see below. Detailed derivation and discussion can be found in Appendix A.4.2.

$$
\begin{cases}
\text{Mutual Information Maximization:} & I(l_i; \boldsymbol{c}_{i,t}, \boldsymbol{s}_{t-1:t-\tau}, a_{t-1:t-\tau}) \\
\text{Mutual Information Minimization:} & \sum_{j \neq i} I(l_i; \boldsymbol{c}_{j,t}, l_j, \boldsymbol{s}_{t-1:t-\tau}, a_{t-1:t-\tau})
\end{cases}
\tag{5}
$$

$$
l_{\text{mi}} = \mathbb{E}\left[\sum_{t=1}^{T}\sum_{i=1}^{m} -\left(I(l_i; \boldsymbol{c}_{i,t}, \boldsymbol{s}_{t-1:t-\tau}, a_{t-1:t-\tau}) - \sum_{j \neq i} I(l_i; \boldsymbol{c}_{j,t}, l_j, \boldsymbol{s}_{t-1:t-\tau}, a_{t-1:t-\tau})\right)\right].
$$

**Enforcing sparse interactions.** Real-world causal systems are usually sparse, with most variables not directly influencing each other (Scholkopf et al., 2021; Zhang & Hyvärinen, 2009). To reflect this characteristic and facilitate learning and identification, we incorporate learnable masks into the world model. We assume that the observation decoding, reward decoding, and continuous signal decoding use only a subset of latents, characterized by binary masks $m_o, m_r, m_c$, respectively. To alleviate the shrinkage effect of L1 loss and prevent excessive sparsity, which could result in information loss during the early stages, we apply a modified gating mask from Rajamanoharan et al.

(2024) and adaptive L1 loss that dynamically controls the sparsity during the optimization process, accounting for the sparsity ratio of the latent variables, with the corresponding loss function given below (see implementation details in Appendix A.5.2):

$$l_{\text{spar}} = \sum_{t=1}^{T} \sum_{i} L_1(m_i) \cdot \mathbf{1}_{\text{ratio}(m_i) < \text{threshold}}.$$

Therefore, the total objective for learning the composable world model is expressed as a weighted summation:

$$l_{\text{total}} = \alpha l_{\text{rep}} + \lambda l_{\text{trans}} + \beta l_{\text{mi}} + \gamma l_{\text{spar}}. \tag{6}$$

**Compact states for policy learning.** We use a standard soft actor-critic structure as our policy module, following DreamerV3 (Hafner et al., 2023). The task embedding $z$, learned from the world model, is used as a condition in both the actor and critic networks to accommodate the multi-task setting. Instead of using all latent states $s_t$, only the compact reward-relevant states, selected via the reward mask $m_r$, are fed into the actor and critic networks. The policy module and world model are optimized alternately, with the world model utilizing the updated policy module to gather new interactions (see Figure 2b).

**Quick adaptation to tasks of new compositions.** Synthesis effect might happen in the test time. For instance, the action *open* within the *verb* component can have different meanings when combined with objects that have distinct affordances, such as *door* and *window*. Instead of standard fine-tuning or retraining on all parameters of the world model, we only require minor adjustments to dynamics-related parts. This is feasible because the individual components have already been learned in previous tasks—it is only the interactions among them that remain unclear. Hence, we fine-tune the task encoder, representation model, transition model, decoding masks, and policy module to account for the synthesis effect of these recombinations while keeping others fixed.

## 3 EXPERIMENTS

In this section, we address the following questions:

- How effectively can our learning framework identify language-controlled components in known environments?
- How accurately can the learned world model predict language-controlled components in new environments with novel combinations?
- Can our framework enhance RL training and generalization in real-world applications?
- Are the learned language-controlled components interpretable?

We aim to answer these questions through a series of experiments, including tests on synthetic data where we have access to ground truth latent states, as well as real-world robotic environments involving a collection of manipulation tasks.

### 3.1 SYNTHETIC DATA

To validate the accuracy of our language-controlled composable component identification, we conduct a simulation study based on the assumed data generation process described in Section 2.1. We initialize three language-controlled components $\{c_1, c_2, c_3\}$, each independently controlled by three groups of discrete language tokens (where language component $l_i$ takes $n_{c_i} + 1$ values). We split all possible combinations into a training set for the *i.i.d* component identification test and a test set for the *o.o.d* component prediction test. Following the experimental setting of block-wise identification in Von Kügelgen et al. (2021); Liu et al. (2023), we compute the coefficient of determination $R^2$ between the estimated components and the ground truth.

We compare the identification accuracy of our method with other causal representation learning approaches, such as iVAE (Khemakhem et al., 2019), which does not incorporate temporal information, TCL (Hyvarinen & Morioka, 2016), and TDRL and NCTRL (Yao et al., 2022; Song et al.,

2024), which use the independence of the Jacobian matrix to identify dimension-wise temporal dependencies. Additionally, we evaluate our method against state-of-the-art world models, including DreamerV3 (Hafner et al., 2023) and TD-MPC2 (Hansen et al., 2023), which do not account for causal representation learning or generalization optimization.

### 3.1.1 COMPONENT IDENTIFICATION IN KNOWN TASKS

WM3C effectively identifies language-controlled components in known tasks. The plot on the left of Figure 3 shows that our method (WM3C) accurately captures the underlying latent structure of the data, with diagonal $R^2 > 0.9$ and off-diagonal values around $\sim 0.1$. The middle figure plots the average $R^2$ values of several baseline models against training steps, where WM3C significantly outperforms the other models, demonstrating both higher accuracy and minimal standard deviation, indicating consistent performance.

We attribute the inferior performance of other methods to differences in model priors and optimization objectives. For example, iVAE uses a standard domain index rather than language components, which limits its ability to identify and utilize composable elements effectively. TDRL, which relies on the independent noise assumption to capture conditional independence among latents, fails to recognize the composable components tied to language. Although NCTRL models domain shifts through an underlying hidden Markov model, the prior is too simplistic to capture complex latent structure changes across domains. Both DreamerV3 and TD-MPC2 focus on fitting models to reconstruct observations, resulting in an entangled representation space that does not prioritize identifying distinct latent components.

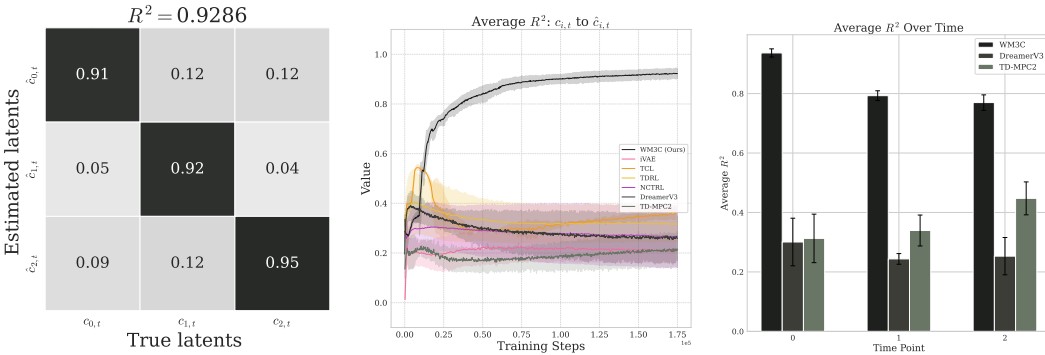

Figure 3: Language-controlled composable components identification results on known tasks and novel tasks. Left: Coefficient of determination ($R^2$) using kernel ridge regression to regress estimated latents on true latents. Middle: Average $R^2$ over the three language-controlled composable components during training, regressing estimated latents on true latents (the shaded areas represent the standard deviation across three runs). Right: Average $R^2$ of model imagination over time during test-time on unseen tasks, where latents are new combinations of known composable components. All results are reported across three runs with different seeds.

### 3.1.2 COMPONENT PREDICTION IN NOVEL TASKS

We evaluate the generalization of the learned model by comparing the latent rollouts generated by the world model (through imagination without consecutive observation) with the true latent in unseen tasks. Here, the unseen tasks are new combinations of language-controlled composable components from the known tasks. In the plot on the right of Figure 3, we see that WM3C consistently maintains the highest $R^2$ values over different time points, reinforcing the effectiveness observed in the middle figure. It is important to note that this evaluation is based on imagination (latent rollouts) in an unseen environment without observations, which highlights WM3C's ability to generalize through learning composable components in the world model.

## 3.2 ROBOT MANIPULATION

To answer whether our framework facilitates RL training and generalization in real-world applications, we conduct experiments in the robotic simulation environment, Meta-world (Yu et al., 2019). It is a benchmark suite of robotic manipulation environments, where each task is paired with a corresponding language description. Assuming language components *verb* and *object* in the data generation process, we take 18 training tasks that follow this structure and 9 test tasks that either are new combinations of known language components or mixtures of known and unknown language components. Models are trained in a multitask setting, including all 18 training tasks, for a total of $5M$ steps, while the adaptation is performed on each test task for $250K$ steps. We compare the training and adaptation efficiency of WM3C (MAE) and WM3C (CNN) with DreamerV3 and visual-based Multi-task SAC (MT-SAC). Additional training and test details are provided in Appendix A.5.2.

### 3.2.1 TRAINING AND ADAPTATION

**Training efficiency.** In Figure 4, we report the average success rate across all tasks during the training process, along with the learning curves of 9 tasks. By learning a composable world model guided by language, our WM3C demonstrates significantly better data efficiency and performance compared to DreamerV3 and MT-SAC, with further improvements achieved through masked image modeling. Additionally, in tasks such as *box-close* and *door-unlock*, where DreamerV3 struggles in learning meaningful representation, our WM3C framework consistently exhibits rapid policy learning. This highlights the WM3C's ability to handle diverse task dynamics and confirms its advantage in learning composable world models as a causal system compared to conventional world models. We provide an ablation study in Appendix A.7.2 to investigate the effects of the integrated modules.

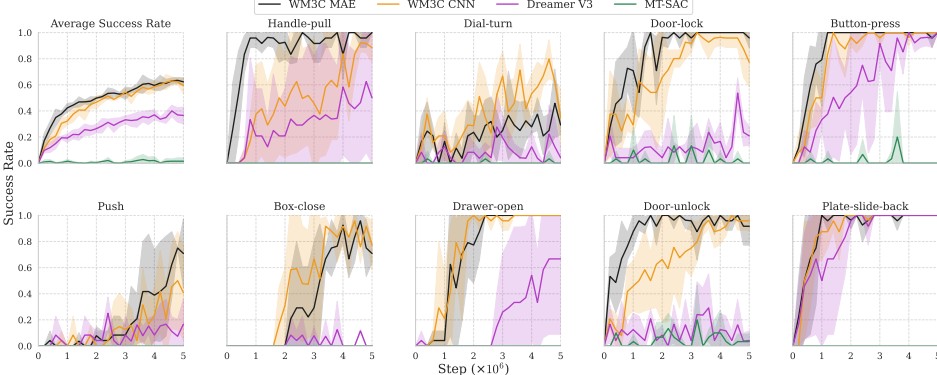

Figure 4: Learning curves on Meta-world: Average success rate and 9 specific task success rates. Results are reported with three seed runs and 10 episodes for each task evaluation.

**Adaptation efficiency.** To evaluate the generalization advantage of the composable world model, we test its adaptation ability on new, unseen tasks, including both the compositions of known components and compositions involving both known and unknown components (See adaptation details in Appendix A.5.2). In Figure 5, we see that in the first 7 tasks that recombine learned components, the WM3C variants, particularly WM3C MAE, consistently achieve higher success rates compared to full-parameter tuned DreamerV3. This aligns with our assumption that after learning composable components, adaptation should focus on learning compositional dynamics. For tasks *coffe-button* (coffee button is unknown, press is known) and *handle-press-side* (handle is known, press-side is unknown), WM3C performs better in one case and worse in both cases, suggesting that while dynamics module adaptation alone may not always suffice for novel components, it can be surprisingly effective in certain similar scenarios.

### 3.2.2 INTERVENTION IN THE LANGUAGE-CONTROLLED COMPONENTS

In this section, we interpret the language-controlled components identified in Meta-world by intervening corresponding latent components and see how this affects the reconstruction. Although not all identifiability conditions can be easily met in practical applications, our estimated latent states

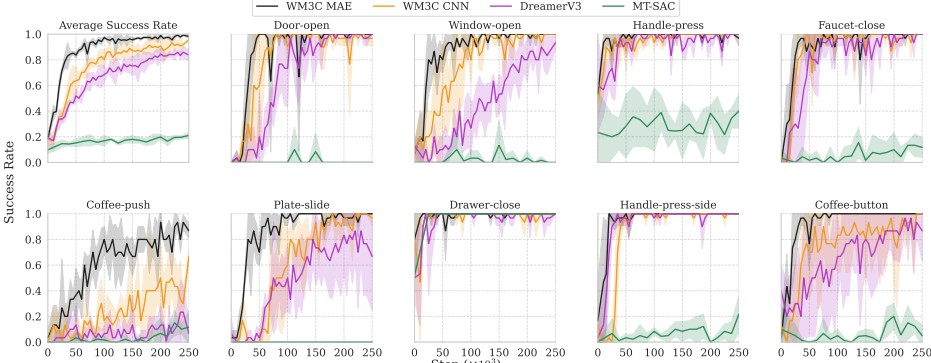

Figure 5: Adaptation curves on 9 unseen tasks, including 7 tasks that are recombination of known components and 2 tasks have unknown components (*handle-press-side* and *coffee-button*). DreamerV3 and MT-SAC are full-parameter finetuned while WM3C is only tuned with the dynamics module. Results are reported with three seed runs and 10 episodes for each task evaluation.

are still reasonable and meaningful (see Figure 6). We find that intervention on the *verb* controlled component does not affect the object in the image but causes the end effector of the robot arm and its adjacent joint to disappear, as expected for the executor of different verbs. On the other hand, intervention on the *object* does not affect the robot arm, but changes the appearance of objects significantly. This demonstrates that our model successfully learns modular and compositional representations aligned with the semantic meaning of language components, enhancing both interpretability and efficient task manipulation. Additional examples can be found in Appendix A.7.3.

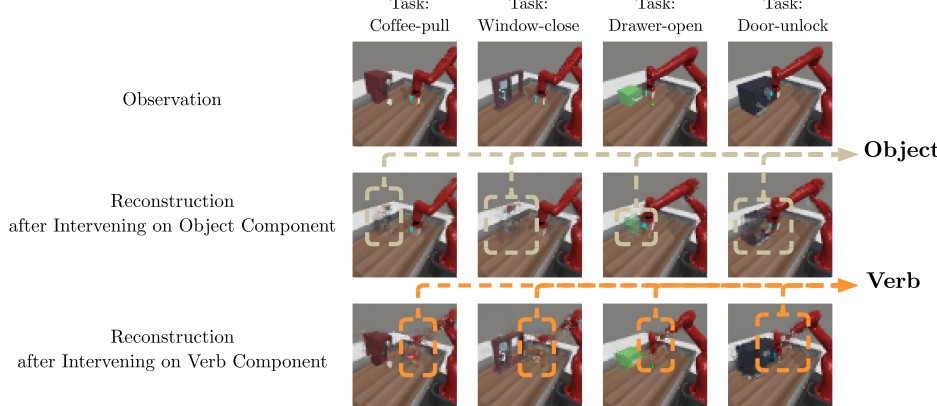

Figure 6: Intervention on the language-controlled components. The first row is the original image observation WM3C receives, the second row is the reconstruction of WM3C's observation decoder after intervening on its *object* latent component and the third row is the reconstruction of WM3C's observation decoder after intervening on its *verb* latent component.

# 4 CONCLUSION

In this work, we introduced World Modeling with Compositional Causal Components (WM3C), a framework aimed at improving generalization in reinforcement learning by utilizing compositional causal dynamics. By integrating modalities like language and offering theoretical guarantees for identifying distinct causal components, WM3C enhances adaptability to unseen environments. Extensive experiments on synthetic data and real-world robotic tasks show WM3C outperforms existing methods in uncovering underlying processes and improving policy learning. Future research can extend WM3C to offline learning and explore scalable modularity and sparsity constraints, considering the large number of language components involved.

## ACKNOWLEDGMENTS

The authors extend their sincere gratitude to the anonymous reviewers for their insightful feedback and constructive suggestions. Biwei Huang would like to acknowledge the support of the National Science Foundation (Grant No. DMS-2428058).

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

# A APPENDIX

## A.1 RELATED WORK

**Generalization in Reinforcement Learning.** Generalization in reinforcement learning (RL) remains a core challenge, particularly when agents need to adapt to unseen task variations or environments with minimal additional training. Early approaches like Meta-RL and invariant representation learning (Lee et al., 2019; Hansen & Wang, 2020; Yuan et al., 2022; Nair et al., 2022) improved adaptability by learning task-agnostic policies that can be efficiently reused across multiple domains. In addition, Model-based approaches, particularly world models (Ha & Schmidhuber, 2018; Hafner et al., 2023; Hansen et al., 2023), have also demonstrated success in generalization by building latent representations of the environment that allow for planning and imagination-based learning. However, these approaches often ignore the compositional nature of tasks, limiting their ability to generalize across tasks that share modular components. In contrast, our work leverages language as a natural guide to identify and control composable task components for a better environment modeling.

**Causal Representation Learning.** Causal representation learning has been a pivotal development in machine learning, focusing on uncovering the underlying causal mechanisms within data (Schölkopf et al., 2021). Prior methods such as nonlinear ICA (Zhang & Hyvarinen, 2012; Hyvärinen & Pajunen, 1999) and temporal causal representation learning (Yao et al., 2021; 2022; Song et al., 2024) have sought to identify causal relationships in latent variables using priors such as independent noise conditions and auxiliary variables. However, these approaches often focus on dimension-wise identifiability, which can be difficult to scale in real-world applications with complex causal dynamics. Our framework advances this line of work by introducing block-wise identifiability across multiple coexisting language components, which establishes a connection to a compositional modality. This further enables the identification of language-controlled composable components as distinct blocks, improving scalability and generalization.

## A.2 PROOF OF PROPOSITION 1 (LANGUAGE-CONTROLLED COMPONENTS)

Here we present the proof for identifiability of the language-controlled composable components. The content is arranged as, firstly deriving the relationship between the estimated latent and ground truth latent variables, then proving that under some language-controlled composable components can be uniquely and correctly identified. The identifiability theory is suitable for the system including arbitrary number of language components and language-controlled latent components. For simplicity, we present the proof with a system having two language components and it can be easily extended to arbitrary number of language components.

We first give the definitions of d-separation, global Markov condition, faithfulness assumption, which are used in the proof (Spirtes et al., 2001; Glymour et al., 2019; Pearl, 2009).

**Definition 1** (d-separation). *In a directed acyclic graph (DAG), a path between two nodes $X$ and $Y$ is said to be blocked (or d-separated) by a set of nodes $Z$ if and only if:*

*For any node on the path:*

- *If the path includes a chain $X \to M \to Y$ or a fork $X \leftarrow M \to Y$, the path is blocked if $M \in Z$.*

- *If the path includes a collider $X \to M \leftarrow Y$, the path is blocked unless $M \in Z$ or any of its descendants are in $Z$.*

**Definition 2** (Global Markov Condition). *The distribution $p$ over a set of variables $V$ satisfies the global Markov property on a directed acyclic graph (DAG) $G$ if for any partition $(X, Z, Y)$ of the variables such that $Z$ d-separates $X$ from $Y$ in $G$, we have:*

$$p(X, Y \mid Z) = p(X \mid Z)p(Y \mid Z)$$

**Definition 3** (Faithfulness Assumption). *A distribution $p$ over a set of variables $V$ is faithful to a directed acyclic graph $G$ if all and only the conditional independencies in $p$ are implied by the global Markov condition on $G$.*

This implies that if two variables are conditionally independent in $p$, this must be reflected by d-separation in $G$, and there are no additional independencies beyond those imposed by the graph structure.

**Proposition 1** (Language-Controlled Components). *Under the assumption that the graphical representation of the environment model is Markov and faithful to the data, $c_{i,t}$ is a minimal subset of state dimensions that are directly controlled by the language component $l_i$ and $s_{j,t} \in c_{i,t}$ if and only if $s_{j,t} \not\perp\!\!\!\perp l_i \mid a_{t-1:t}, s_{t-1}$, and $s_{j,t} \perp\!\!\!\perp \{l_k\}_{k \neq i} \mid l_i, a_{t-1:t}, s_{t-1}$.*

*Proof.* The proof is done with twice contradiction as follows:

**Step 1:** We first show that if $s_{j,t} \in c_{i,t}$, then $s_{j,t} \not\perp\!\!\!\perp l_i \mid a_{t-1:t}, s_{t-1}$.

We prove this by contradiction. Assume $s_{j,t} \perp\!\!\!\perp l_i \mid a_{t-1:t}, s_{t-1}$, meaning that $s_{j,t}$ is independent of $l_i$ given the previous actions $a_{t-1:t}$ and state dimensions $s_{t-1}$. According to the faithfulness assumption, this implies that there is no directed path from $l_i$ to $s_{j,t}$ in the graphical model, contradicting the assumption that $s_{j,t}$ is part of $c_{i,t}$, which consists of state dimensions controlled by $l_i$. Thus, by contradiction, it follows that $s_{j,t} \not\perp\!\!\!\perp l_i \mid a_{t-1:t}, s_{t-1}$, meaning that $s_{j,t}$ has a directed path from $l_i$, and is thereby controlled by $l_i$.

**Step 2:** Next, we show that $s_{j,t} \perp\!\!\!\perp \{l_k\}_{k \neq i} \mid l_i, a_{t-1:t}, s_{t-1}$.

Again, suppose by contradiction that $s_{j,t} \not\perp\!\!\!\perp \{l_k\}_{k \neq i} \mid l_i, a_{t-1:t}, s_{t-1}$, implying that $s_{j,t}$ is dependent on another language component $l_k$ (for some $k \neq i$) even after conditioning on $l_i$, the action sequence, and the previous state. According to the faithfulness assumption, this would mean there is a direct path from $l_k$ to $s_{j,t}$. However, this contradicts the proposition's condition that $s_{j,t} \in c_{i,t}$, which requires that $s_{j,t}$ is controlled specifically by $l_i$ and not influenced by any other $l_k$ for $k \neq i$. Therefore, by contradiction, it must be true that $s_{j,t} \perp\!\!\!\perp \{l_k\}_{k \neq i} \mid l_i, a_{t-1:t}, s_{t-1}$.

From the two steps, we have shown that $s_{j,t} \in c_{i,t}$ if and only if $s_{j,t} \not\perp\!\!\!\perp l_i \mid a_{t-1:t}, s_{t-1}$ and $s_{j,t} \perp\!\!\!\perp \{l_k\}_{k \neq i} \mid l_i, a_{t-1:t}, s_{t-1}$. Thus, $c_{i,t}$ is a minimal subset of state dimensions controlled by the language component $l_i$. $\blacksquare$

## A.3 PROOF OF IDENTIFYING LANGUAGE-CONTROLLED LATENT COMPONENTS

Consider the data generation process described by Equations (1) and (2). To simplify the notation, we demonstrate the proof for a system with two language components, $\{l_1, l_2\}$, and their corresponding language-controlled latent components, $\{c_{1,t}, c_{2,t}\}$. The proof generalizes to any number of language components and associated latent variables.

$$[o_t, r_t] = g(s_t, \epsilon_t) \quad s_t = (c_{1,t}, \ldots, c_{m,t}) \tag{7}$$

$$c_t^i \sim p(c_t^i | l_i, s_{t-1}, a_{t-1}) \quad \text{for } i = 1, \ldots, n \tag{8}$$

We begin by matching the observation distributions:

$$p(o_t, r_t \mid l_1, l_2, s_{t-1}, a_{t-1}) = p(\hat{o}_t, \hat{r}_t \mid l_1, l_2, \hat{s}_{t-1}, a_{t-1}), \tag{9}$$

$$p(g(s_t, \epsilon_t) \mid l_1, l_2, s_{t-1}, a_{t-1}) = p(\hat{g}(\hat{s}_t, \hat{\epsilon}_t) \mid l_1, l_2, \hat{s}_{t-1}, a_{t-1}). \tag{10}$$

The proof of the elimation of noise can be referred to Khemakhem et al. (2020). Assuming that the observation function $g$ is invertible and differentiable, and applying the change of variables formula, we have:

$$p(s_t \mid l_1, l_2, s_{t-1}, a_{t-1}) = p(\hat{s}_t \mid l_1, l_2, \hat{s}_{t-1}, a_{t-1}) \left| \det J_h^{-1} \right|, \tag{11}$$

where $h = g^{-1} \circ \hat{g}$ is the invertible transformation from the estimated latent variables $\hat{s}_t$ to the true latent variables $s_t$, and $J_h$ is the Jacobian of $h$.

Taking the logarithm of both sides:

$$\log p(\boldsymbol{s}_t \mid l_1, l_2, \boldsymbol{s}_{t-1}, a_{t-1}) = \log p(\hat{\boldsymbol{s}}_t \mid l_1, l_2, \hat{\boldsymbol{s}}_{t-1}, a_{t-1}) + \log \left| \det J_h^{-1} \right|. \tag{12}$$

Under Assumption 3, given the language token $l_1$ and $l_2$, the previous state $\boldsymbol{s}_{t-1}$ and the previous action $a_{t-1}$, each latent variable $\boldsymbol{s}_{i,t}$ is independent of the other latent variables. Thus, the log-density of $\boldsymbol{s}_t$ given $l_1, l_2$ can be decomposed as:

$$\log p(\boldsymbol{s}_t \mid l_1, l_2, \boldsymbol{s}_{t-1}, a_{t-1}) = \sum_{i=1}^{n} \log p(s_{i,t} \mid l_1, l_2, \boldsymbol{s}_{t-1}, a_{t-1}), \tag{13}$$

and similarly for the estimated latent variables:

$$\log p(\hat{\boldsymbol{s}}_t \mid l_1, l_2, \hat{\boldsymbol{s}}_{t-1}, a_{t-1}) = \sum_{i=1}^{n} \log p(\hat{s}_{i,t} \mid l_1, l_2, \hat{\boldsymbol{s}}_{t-1}, a_{t-1}). \tag{14}$$

Substituting these into the previous equation:

$$\sum_{i=1}^{n} \log p(s_{i,t} \mid l_1, l_2, \boldsymbol{s}_{t-1}, a_{t-1}) = \sum_{i=1}^{n} \log p(\hat{s}_{i,t} \mid l_1, l_2, \hat{\boldsymbol{s}}_{t-1}, a_{t-1}) + \log \left| \det J_h^{-1} \right|. \tag{15}$$

Here, we first identify $\boldsymbol{c}_1$, the $l_1$ controlled latent component. We use dimension index $\{1, ..., n_{\boldsymbol{c}_1}\}$ to indicate the latent variables belonging to $\boldsymbol{c}_1$, and dimension idex $\{n_{\boldsymbol{c}_1} + 1, ..., n\}$ to indicate the latent variables belonging to $\boldsymbol{c}_2$, where $n$ represents the total dimensions of the latent space. We further simplify the notation that $q_i(s_{i,t}, l_1, l_2) := \log p(s_{i,t} \mid l_1, l_2, \boldsymbol{s}_{t-1}, a_{t-1})$ and $\hat{q}_i(\hat{s}_{i,t}, l_1, l_2) := \log p(\hat{s}_{i,t} \mid l_1, l_2, \hat{\boldsymbol{s}}_{t-1}, a_{t-1})$.

We begin by taking the derivative with the estimated latent variable $\hat{s}_{j,t}$ where $j \in \{n_{\boldsymbol{c}_1} + 1, ..., n\}$.

$$\sum_{i=1}^{n} \frac{\partial q_i(s_{i,t}, l_1, l_2)}{\partial s_{i,t}} \frac{\partial s_{i,t}}{\partial \hat{s}_{j,t}} = \frac{\partial \hat{q}_j(\hat{s}_{j,t}, l_1, l_2)}{\partial \hat{s}_{j,t}} + \frac{\log \left| \det J_h^{-1} \right|}{\partial \hat{s}_{j,t}} \tag{16}$$

Here we build the connection between the dimensions in the true $l_1$ controlled component $s_{i,t}$ and the estimated latent in the $l_2$ controlled component, $s_{j,t}$. Although it is helpful in the sense of identification of $\boldsymbol{c}_1$, it is challenging in estimation because we do not have knowledge about the invertible function, $h$, making the estimation intractable.

Fortunately, if we have multiple values of $l_1$, we can leverage them to make the estimation tractable by constructing the difference terms. We assume $l_1$'s value can be taken from $\{l_{1,0}, ..., l_{1,k}\}$.

$$\sum_{i=1}^{n} \frac{\partial q_i(s_{i,t}, l_{1,k}, l_2)}{\partial s_{i,t}} \frac{\partial s_{i,t}}{\partial \hat{s}_{j,t}} - \frac{\partial q_i(s_{i,t}, l_{1,0}, l_2)}{\partial s_{i,t}} \frac{\partial s_{i,t}}{\partial \hat{s}_{j,t}} = \frac{\partial \hat{q}_j(\hat{s}_{j,t}, l_{1,k}, l_2)}{\partial \hat{s}_{j,t}} - \frac{\partial \hat{q}_j(\hat{s}_{j,t}, l_{1,0}, l_2)}{\partial \hat{s}_{j,t}} \tag{17}$$

Recall that the independent assumption that the latent component controlled by $l_2$ are not controlled by $l_1$. $\hat{q}_j(\hat{s}_{j,t}, l_{1,k}, l_2)$ does not change when only $l_1$ changes. However, this also adds some constraints and requirements on the number of tasks that combined $l_1$ and $l_2$, which we are going to discuss later. Here we can remove the $l_2$ from the conditions since according to the proposition of $s_{i,t} \in \boldsymbol{c}_{1,t}$, it is independent from $l_2$ given $l_1$, $\boldsymbol{s}_{t-1}$ and $a_{t-1}$.

$$\sum_{i=1}^{n_{\boldsymbol{c}_1}} \left( \frac{\partial q_i(s_{i,t}, l_{1,k})}{\partial s_{i,t}} - \frac{\partial q_i(s_{i,t}, l_{1,0})}{\partial s_{i,t}} \right) \frac{\partial s_{i,t}}{\partial \hat{s}_{j,t}} = 0 \tag{18}$$

Suppose we have $n_{\boldsymbol{c}_1}$ values in $l_1$, from $l_{1,0}$ to $l_{1,k}$, we can construct a linear system as follows:

$$\begin{bmatrix} \frac{\partial q_1(s_{1,t},l_{1,1})}{\partial s_{1,t}} - \frac{\partial q_1(s_{1,t},l_{1,0})}{\partial s_{1,t}} & \cdots & \frac{\partial q_{n_{c_1}}(s_{n_{c_1},t},l_{1,1})}{\partial s_{n_{c_1},t}} - \frac{\partial q_{n_{c_1}}(s_{n_{c_1},t},l_{1,0})}{\partial s_{n_{c_1},t}} \\ \vdots & & \vdots \\ \frac{\partial q_1(s_{1,t},l_{1,k})}{\partial s_{1,t}} - \frac{\partial q_1(s_{1,t},l_{1,0})}{\partial s_{1,t}} & \cdots & \frac{\partial q_{n_{c_1}}(s_{n_{c_1},t},l_{1,k})}{\partial s_{n,t}} - \frac{\partial q_{n_{c_1}}(s_{n_{c_1},t},l_{1,0})}{\partial s_{n_{c_1},t}} \end{bmatrix} \begin{bmatrix} \frac{\partial s_{1,t}}{\partial \hat{s}_{j,t}} \\ \vdots \\ \frac{\partial s_{n_{c_1},t}}{\partial \hat{s}_{j,t}} \end{bmatrix} = 0. \quad (19)$$

To achieve $\frac{\partial s_{i,t}}{\partial \hat{s}_{j,t}} = 0$ for any $i \in \{1, \ldots, n_{c_1}\}$ and $j \in \{n_{c_1} + 1, \ldots, n\}$, the left matrix has to be full rank. This further implies that if the cardinality of $l_1$'s range ($k$) is larger than $n_{c_1} + 1$, $\frac{\partial c_1}{\partial \hat{c}_2} = 0$, as well as $\frac{\partial c_1}{\partial \hat{\bar{c}}_1} = 0$.

Next, we turn to the Jacobian matrix of $h$ that describes the relationship between the true latent variables $s_t$ and the estimated latent variables $\hat{s}_t$:

$$J_h = \begin{bmatrix} \frac{\partial c_{1,t}}{\partial \hat{c}_{1,t}} & \frac{\partial c_{1,t}}{\partial \hat{c}_{2,t}} \\ \frac{\partial c_{2,t}}{\partial \hat{c}_{1,t}} & \frac{\partial c_{2,t}}{\partial \hat{c}_{2,t}} \end{bmatrix}. \tag{20}$$

By assuming the transformation $h$ is invertible, the Jacobian matrix $J_h$ is full rank. We can safely meet conclude that, $\frac{\partial s_{c_1}}{\partial \hat{s}_{c_2}} = 0$, and the non-zero entries can only appear in $\frac{\partial s_{c_1}}{\partial \hat{s}_{c_1}}$. The $c_2$ here can be further extended to the concatenation of components that are not controlled by $l_1$, namely $\bar{c}_1$ in the cases of having language components more than 2.

After proving the number of $l_1$ for identifying $c_1$, it is straightforward to know the number of $l_i$ needs and the number of tasks (combinations of $l_i$) for identifying all the language-controlled components.

We can use the same procedure to prove the number of $l_2$ needed for identifying $c_1$ is the same as proving the identification of $c_1$, which requires $n_{c_2} + 1$ values of $l_2$.

Below, we discuss two estimation procedures for identifying all the language-controlled components, related to the tradeoff between practicality and minimal number of tasks.

For the case having two language components, suppose we have enough data, e.g. sufficient variability in $l_1$ and $l_2$. We have two ways to identify $c_1$ and $c_2$.

- **One by one:** We can first identify one component with a model and then identify another component with another model in a one-by-one order. The least number of tasks we need is $\sum_{i=1}^{2} n_{c_i} + 1$ for 2 language components and $\sum_{i=1}^{m} n_{c_i} + 1$ for $m$ language components. However, it is limited in practical use since we can not identify all language-controlled components simultaneously in the estimation procedure. In order to meet the requirement in Equation 17, we need to control the $\hat{q}_j(\hat{s}_{j,t}, l_{1,k}, l_2)$ does not change when only $l_{1,k}$. It requires us to have tasks that allow us to control the $l_2$ to be a fixed value when $l_1$ is changing, and vice versa. Then, we can use these tasks to estimate $c_1$ and in turn estimate the $c_2$ with another set of tasks where $l_1$ is fixed and $l_2$ is changing. While it requires the minimal number of tasks and language tokens, the estimation procedure is sophisticated.

- **All in one:** We can learn a model to simultaneously identify and estimate all the components with more needs on variability. To make sure the the $\hat{q}_j(\hat{s}_{j,t}, l_{1,k}, l_2)$ does not change when only $l_{1,k}$, we need enough values of $l_2$ presented with each value of $l_1$. That helps because when $l_1$ is changing, there is also enough $l_2$ values to let us identify $c_2$. It is the same in the case where more than 2 language components evolved, where we want to have enough combinations of the rest language components' values. This would require $\prod_{i=1}^{2} n_{c_i}$ tasks for 2 language components and $\prod_{i=1}^{m} n_{c_i} + 1$ tasks for $m$ language components.

Once the latent variables are identified using Theorem 1, the causal structure among these latent variables can be directly recovered using standard causal discovery methods, such as constraint-based or score-based approaches used in prior work (Yao et al., 2021; 2022; Song et al., 2024). These methods leverage the identified latent space to ensure the underlying causal relationships are correctly estimated, providing a clear path to causal structure identifiability.

### A.4    Objective Derivations

#### A.4.1    Representation and dynamics objectives

We define the extended information bottleneck objective based on Dreamer (Hafner et al., 2019), now conditioning on $L = \{l_1, ..., l_m\}$, as follows:

$$\max I(s_{1:T}; (o_{1:T}, r_{1:T})|a_{1:T}, L) - \beta I(s_{1:T}, i_{1:T}|a_{1:T}, L),$$

where $\beta$ is a scalar and $i_t$ are dataset indices that determine the observations $p(o_t|i_t) = \delta(o_t - \bar{o}_t)$ as in .

Maximizing this objective encourages the model to reconstruct each image by relying on information extracted at preceding time steps to the extent possible, and only accessing additional information from the current image when necessary.

Lower bounding the first term gives us the objective of optimizing the latent representation.

$$I(\boldsymbol{s}_{1:T}; (o_{1:T}, r_{1:T})|a_{1:T}, L)$$

$$= \mathbb{E}_{p(o_{1:T}, r_{1:T}, \boldsymbol{s}_{1:T}, a_{1:T}, L)} \left( \sum_t \ln p(o_{1:T}, r_{1:T}|\boldsymbol{s}_{1:T}, a_{1:T}, L) - \ln p(o_{1:T}, r_{1:T}|a_{1:T}, L) \right)$$

$$\overset{\pm}{=} \mathbb{E} \left( \sum_t \ln p(o_{1:T}, r_{1:T}|\boldsymbol{s}_{1:T}, a_{1:T}, L) \right)$$

$$\geq \mathbb{E} \left( \sum_t \ln p(o_{1:T}, r_{1:T}|\boldsymbol{s}_{1:T}, a_{1:T}, L) \right) - \mathrm{KL} \left( p(o_{1:T}, r_{1:T}|\boldsymbol{s}_{1:T}, a_{1:T}, L) \parallel \prod_t q(o_t|\boldsymbol{s}_t)q(r_t|\boldsymbol{s}_t) \right)$$

$$= \mathbb{E} \left( \sum_t \ln q(o_t|\boldsymbol{s}_t) + \ln q(r_t|\boldsymbol{s}_t) \right).$$

Upper bounding the second term gives us the objective of optimizing the latent transition dynamics.

$$I(\boldsymbol{s}_{1:T}; i_{1:T}|a_{1:T}, L)$$

$$= \mathbb{E}_{p(o_{1:T}, r_{1:T}, \boldsymbol{s}_{1:T}, a_{1:T}, i_{1:T}, L)} \left( \sum_t \ln p(\boldsymbol{s}_t|s_{t-1}, a_{t-1}, i_t, L) - \ln p(\boldsymbol{s}_t|\boldsymbol{s}_{t-1}, a_{t-1}, L) \right)$$

$$= \mathbb{E} \left( \sum_t \ln p(\boldsymbol{s}_t|\boldsymbol{s}_{t-1}, a_{t-1}, o_t, L) - \ln p(\boldsymbol{s}_t|\boldsymbol{s}_{t-1}, a_{t-1}, l_j) \right)$$

$$\leq \mathbb{E} \left( \sum_t \ln p(\boldsymbol{s}_t|\boldsymbol{s}_{t-1}, a_{t-1}, o_t, L) - \ln q(\boldsymbol{s}_t|\boldsymbol{s}_{t-1}, a_{t-1}, L) \right)$$

$$= \mathbb{E} \left( \sum_t \mathrm{KL} \left( p(\boldsymbol{s}_t|\boldsymbol{s}_{t-1}, a_{t-1}, o_t, L) \parallel q(\boldsymbol{s}_t|\boldsymbol{s}_{t-1}, a_{t-1}, L) \right) \right)$$

This lower bounds the objective. It can be further decomposed to the summation of independent KL terms according to the environment model we assume in Section. 2.1 and the Proposition A.2 of the language-controlled component.

$$I(\boldsymbol{s}_{1:T}; i_{1:T}|a_{1:T}, L)$$

$$= \mathbb{E}\left(\sum_t \text{KL}\left(p(\boldsymbol{s}_t|\boldsymbol{s}_{t-1}, a_{t-1}, o_t, L) \parallel q(\boldsymbol{s}_t|\boldsymbol{s}_{t-1}, a_{t-1}, L))\right)\right)$$

$$= \mathbb{E}\left(\sum_t \sum_i^m \text{KL}\left(p(\boldsymbol{c}_{i,t}|\boldsymbol{s}^{t-1}, a_{t-1}, o_t, l_i) \parallel q(\boldsymbol{c}_{i,t}|\boldsymbol{s}_{t-1}, a_{t-1}, l_i))\right)\right)$$

### A.4.2 MUTUAL INFORMATION CONSTRAINT DERIVATIONS

In order to enhance the conditional independence in the latent space, we also adopt the mutual information constraints, which are as follows:

$$I(l_i; \boldsymbol{c}_{i,t} \mid \boldsymbol{s}_{t-1:t-\tau}, a_{t-1:t-\tau}) - \sum_{j \neq i} I(l_i; \boldsymbol{c}_{j,t} \mid l_j, \boldsymbol{s}_{t-1:t-\tau}, a_{t-1:t-\tau})$$

It is used to characterize the language-controlled component $\boldsymbol{c}_i$, by enhancing the dependence between $\boldsymbol{c}_i$ and $l_i$, given $a_{t-1:t-\tau}, \boldsymbol{s}_{t-1:t-\tau}$, and enhancing the independence between $l_i$ and each $\boldsymbol{c}_j$ conditioning on corresponding $l_j, \boldsymbol{s}_{t-1:t-\tau}, a_{t-1:t-\tau}$. The $\tau$ here refers to the length of history that considering conditioning on, which is normally 1 when assumed markov assumption. By applying the chain rule of mutual information for the first term and second term, we can have the following:

$$I(l_i; \boldsymbol{c}_{i,t} \mid \boldsymbol{s}_{t-1:t-\tau}, a_{t-1:t-\tau}) = I(l_i; \boldsymbol{c}_{i,t}\boldsymbol{s}_{t-1:t-\tau}, a_{t-1:t-\tau}) - I(l_i; \boldsymbol{s}_{t-1:t-\tau}, a_{t-1:t-\tau})$$
$$I(l_i; \boldsymbol{c}_{j,t} \mid l_j, \boldsymbol{s}_{t-1:t-\tau}, a_{t-1:t-\tau}) = I(l_i; \boldsymbol{c}_{j,t}, l_j, \boldsymbol{s}_{t-1:t-\tau}, a_{t-1:t-\tau}) - I(l_i; l_j, \boldsymbol{s}_{t-1:t-\tau}, a_{t-1:t-\tau})$$

In order to simultaneously optimize the representation and dynamics objectives, we apply the stop gradient to $\boldsymbol{s}_{t-1}$, which converts the optimization of the conditional mutual information to the joint mutual information since the second terms in both directions are constant.

$$I(l_i; \boldsymbol{c}_{i,t} \mid \boldsymbol{s}_{t-1:t-\tau}, a_{t-1:t-\tau}) \stackrel{\pm}{=} I(l_i; \boldsymbol{c}_{i,t}\boldsymbol{s}_{t-1:t-\tau}, a_{t-1:t-\tau})$$
$$I(l_i; \boldsymbol{c}_{j,t} \mid l_j, \boldsymbol{s}_{t-1:t-\tau}, a_{t-1:t-\tau}) \stackrel{\pm}{=} I(l_i; \boldsymbol{c}_{j,t}, l_j, \boldsymbol{s}_{t-1:t-\tau}, a_{t-1:t-\tau})$$

Then we obtain the mutual information constraints in 5. For each languauge component $l_i$, we maximize the conditional mutual information between it and its corresponding language-controlled component $\boldsymbol{c}_i$ and minimize the summation of the conditional mutual information between it and the other language-controlled component $\boldsymbol{c}_j$.

$$\sum_{t=1}^{T} \sum_i^{m=1} I(l_i; \boldsymbol{c}_{i,t}, \boldsymbol{s}_{t-1:t-\tau}, a_{t-1:t-\tau}) - \sum_{j \neq i} I(l_i; \boldsymbol{c}_{j,t}, l_j, \boldsymbol{s}_{t-1:t-\tau}, a_{t-1:t-\tau})$$

Concretely, we learn groups of mutual information estimators to estimate these mutual information values by maximizing the Donsker-Varadhan representation Lower Bound with mutual information neural estimator Belghazi et al. (2018), then use these estimators to help constrain the latent space of world model.

### A.5 EXPERIMENT SETTINGS

### A.5.1 NUMERAICAL SIMULATION

In the simulation process, we follow the data generation process in Equation 1 and 2, and identifiability conditions in Thereom 1. While our framework is suitable for any number of language-controlled

components, we use three language components $\{l_1, l_2, l_3\}$ for simplicity. The latent variables $s_t$ have 6 dimensions, where $n_{c_1} = n_{c_2} = n_{c_3} = 2$. We set the number of values in each $l_i$ as 3 and generate all 27 possible combinations of tasks, indicated by $(l_1, l_2, l_3)$, e.g. $(0, 1, 2)$. We take three tasks that are the recombinations of values that appeared in the other tasks as test tasks, and use the rest tasks as training tasks. Even though this makes us cannot have an optimal number of tasks to identify all components simultaneously in a perfect way, the $R^2$ shows that it does not affect a lot. To make sure that different language components have different effects on the transition dynamics, we further embed them using embedding layers of different sizes before incorporating them in the transition functions. At each time step, a one-hot action of dimension 3 is taken. The functions in the data generation process are initialized with MLPs. This setting allows us to take the all-in-one strategy to identify $c_1$, $c_2$ and $c_3$.

### A.5.2 META-WORLD

**Environemnt Details** Meta-World is a benchmark suite of 50 robotic manipulation environments designed for multitask and meta-reinforcement learning, where each task is accompanied by a corresponding language description. To best achieve the identifiability condition, we choose the common language component system that include most tasks, *verb* and *object*. We use these 18 tasks as training tasks and the other 9 tasks that haven't shown in the training tasks but can be either presented as the recombination of language components in the training tasks or tasks that have one known component in the training tasks, as test tasks. The tasks are as follows:

| Set | Task | Language Components (Verb, Object) |
|---|---|---|
| Train | Box-Close | Pick-Place, Cover |
| | Bin-Picking | Pick-Place, Bin |
| | Basketball | Pick-Place, Basketball |
| | Soccer | Push, Soccer |
| | Button-Press | Press, Button |
| | Coffee-Pull | Pull, Mug |
| | Dial-Turn | Open, Dial |
| | Door-Close | Close, Door |
| | Door-Lock | Lock, Door |
| | Door-Unlock | Unlock, Door |
| | Faucet-Open | Open, Faucet |
| | Handle-Pull | Pull, Handle |
| | Push-Back | Pull, Puck |
| | Push | Push, Puck |
| | Plate-Slide-Back | Retrieve, Plate |
| | Sweep | Sweep, Puck |
| | Window-Close | Close, Window |
| | Drawer-Open | Open, Drawer |
| Test | Faucet-Close[†] | Close, Faucet |
| | Coffee-Push[†] | Push, Mug |
| | Handle-Press[†] | Press, Handle |
| | Drawer-Close[†] | Close, Drawer |
| | Window-Open[†] | Open, Window |
| | Door-Open[†] | Open, Door |
| | Plate-Slide[†] | Push, Plate |
| | Handle-Press-Side[*] | Press-Side, Handle |
| | Coffee-Button[*] | Press, Coffee Button |

Table 1: Task descriptions in Meta-World training and test sets. [†]Test tasks that are recombinations of verb-object components from the training set. [*]Test tasks that contain either a novel verb variant (Press-Side) or a novel object (Coffee Button) not present in the training set.

**Model Details** For WM3C CNN, we build upon the JAX implementation of DreamerV3 - small and use the medium-size visual encoder and policy module, to balance the extra parameters introduced by the modifications. For WM3C MAE, we substitute the CNN encoder and decoder in the WM3C

CNN with MAE and keep the others to be the same as WM3C CNN, referring Seo et al. (2023). The details in implementing mutual information constraints and learnable masks are described as below:

- **MI** The mutual information estimators are implemented in JAX, referring MINE (Belghazi et al., 2018). An cosine annealing scheduler as below is used to smoothly adjust the coefficient of mutual information constraints in the objective from zero to the set value to prevent the training instability brought by inaccurate estimation and variances in min-max optimization in the early stages.

$$\text{scale}(t) = \begin{cases} v_{\text{start}} + \frac{1}{2}(v_{\text{end}} - v_{\text{start}})(1 - \cos(\pi t)) & \text{if } v_{\text{end}} > v_{\text{start}} \\ v_{\text{end}} + \frac{1}{2}(v_{\text{start}} - v_{\text{end}})(1 + \cos(\pi t)) & \text{otherwise} \end{cases} \tag{21}$$

- **Mask** We apply a modified gated masks inspired by (Rajamanoharan et al., 2024) to have more flexibility of sparsity and alleviate the shrinkge effect of L1 loss, which potentially affects the exploration efficiency of policy. To accurately control the sparsity rate in the latents and mitigate the ineffective learning because of over-sparse, we also apply cosine annealing scheduler to smoothly increase the sparsity rate threshold from zero, jointly with the adaptive L1 loss. We apply the masks only to the deterministic part and keep the stochastic part of latent to be unchanged. The masking formula and L1 loss are as follows:

$$m \odot \boldsymbol{x} := \mathbf{1}_n \underbrace{\left[ \underbrace{|\boldsymbol{x}| + b_{\text{gate}}}_{m_{\text{gate}}(\boldsymbol{x})} > 0 \right]}_{} \odot \underbrace{(\exp(r_{\text{mag}}) \cdot \boldsymbol{x} + b_{\text{mag}})}_{f_{\text{mag}}(\boldsymbol{x})} \tag{22}$$

$$L_1(m) := \|\text{ReLU}(m_{\text{gate}})\|_1 \tag{23}$$

For DreamerV3, we take the official JAX implementation of DreamerV3 (Hafner et al., 2023) from https://github.com/danijar/dreamerv3, and use the medium version for all experiments.

For visual-based multi-task SAC (MT-SAC), we take the visual-based SAC implementation from https://github.com/KarlXing/RL-Visual-Continuous-Control and modified it to the multi-task SAC by including the contextual embedding of the task language description, as well as the hyperparameters for Meta-world, referring CARE (Sodhani et al., 2021).

**Training Details** In the training stage, we train all the models (WM3C, DreamerV3, MT-SAC) for $5M$ environment steps in total in a multi-task online learning way, which on average for each task is around $280K$ steps.

**Adaptation Details** In the adaptation stage, we fine-tune all the pretrained models (WM3C, DreamerV3, MT-SAC) for 250K steps separately for each test task. The 250K steps can be reduced since for most tasks, the adaptation does not require much change and converges much earlier than that. For WM3C CNN and MAE, we only fine-tune the factorized dynamics module (representation models and transition models), task encoder, decoding masks, and policy module, while the rest of the model is frozen. We improve the threshold of the sparsity rate to encourage a more strict selection of the latent for decoding since for a specific task, the ground truth latent should be less than the latent related to multiple tasks. For DreamerV3 and MT-SAC, we fine-tune all the parameters.

**Hyperparameters** We summarized the important hyperparameters of WM3C CNN as follows: The WM3C MAE is the same as WM3C CNN except the visual encoder and decoder are substituted with vanilla MAE. The important hyperparameters are as follows:

### A.5.3 COMPUTATIONAL RESOURCES

All experiments are conducted a $4 \times$ Nvidia 3090 GPU. Training from scratch on the simulation experiment takes 2 hours for one run, training on the 18 Meta-world tasks takes 8 days for $5M$ steps and each fine-tuning task takes 8 hours for $250K$ steps.

## A.6 EXTENDED DISCUSSIONS

### A.6.1 COMPARISON WITH HIERARCHICAL RL

WM3C and hierarchical reinforcement learning (HRL) (Barto & Mahadevan, 2003; Nachum et al., 2018; Parr & Russell, 1997) share a conceptual similarity in their focus on decomposing tasks into

| Category | Hyperparameter | Value |
|---|---|---|
| General | Batch size | 50 |
| | Batch length | 50 |
| | Train ratio | 500 |
| | Training fill | 5000 |
| | Image size | (64, 64, 3) |
| World Model | Component deterministic size | 512 |
| | Component stochastic size | 16 |
| | Component classes | 32 |
| | CNN depth | 48 |
| | Reward&Cont layers | 2 |
| | Reward&Cont units | 512 |
| | Actor&Critic layers | 3 |
| | Actor&Critic units | 640 |
| | Task embedding dim | 128 |
| | Token embedding dim | 128 |
| | Mask types | [decoder, reward, cont] |
| Training | Training steps | $5 \times 10^6$ |
| | Sparsity rate | 0.25 |
| | Optimize MINE after steps | 0 |
| | Maximize MINE after steps | 1e4 |
| | Minimize MINE after steps | 1e4 |
| Adaptation | Adaptation steps | $2.5 \times 10^5$ |
| | Sparsity rate | 0.35 |
| | Maximize MINE after steps | 0 |
| | Minimize MINE after steps | 0 |

Table 2: Hyperparameters for WM3C CNN implementation in Meta-world.

| Component | Hyperparameter | Value |
|---|---|---|
| MAE Encoder | Patch size | 8 |
| | Embedding dimension | 256 |
| | Encoder depth | 4 |
| | Number of heads | 4 |
| | Mask ratio | 0.75 |
| MAE Decoder | Decoder embedding dim | 256 |
| | Decoder depth | 3 |
| | Decoder heads | 4 |
| | Early convolution | True |
| ViT Parameters | Image size | 8 |
| | Patch size | 1 |
| | Embedding dimension | 128 |
| | Depth | 2 |
| | Number of heads | 4 |
| | Input channels | 256 |

Table 3: MAE and ViT hyperparameters for WM3C MAE in Meta-world. The architecture decouples the visual learning and dynamics learning, and small ViTs are added in the dynamics module to align visual and dynamics information.

smaller, more manageable components. However, their methodologies and applications diverge significantly. HRL typically structures policies into hierarchies, such as high-level managers setting subgoals for low-level controllers. These subgoals are often state-specific and lack explicit modeling of compositionality or causal dynamics among task components.

In contrast, WM3C explicitly learns compositional causal components grounded in the environment's causal structure, enabling robust adaptation to unseen tasks by recombining previously learned components. This distinction is particularly evident in WM3C's use of language as a compositional modality, which aligns causal components with interpretable semantic elements. While HRL might excel in navigating hierarchical tasks, it may struggle with environments requiring flexible recombination of causal dynamics—a strength of WM3C.

### A.6.2 COMPARISON WITH PREVIOUS NON-LINEAR ICA AND CAUSAL REPRESENTATION LEARNING

WM3C builds upon the principles of non-linear independent component analysis (ICA) but departs from prior approaches in two critical ways. First, non-linear ICA methods, such as iVAE (Khemakhem et al., 2019), SlowVAE (Klindt et al., 2020) and TCL (Hyvarinen & Morioka, 2016), focus on identifying latent variables under strong assumptions like independence of noise terms or specific priors on functional forms. These methods are often dimension-wise and fail to scale effectively to complex systems with interdependent latent structures. Furthermore, WM3C considers the case of multiple auxiliary variables (language components) co-existing, while prior results only allow one auxiliary variable and a much simpler graph rather than the graphical model in the POMDP process. WM3C does not assume a parametric expression to acquire identification, enabling a more flexible and practical estimation.

WM3C introduces block-wise identifiability, enabling the identification of language-controlled compositional components rather than individual latent variables. We allow the separate language components connecting to the latents they control rather than having only one auxiliary variable connecting to all the latents, different from previous non-linear ICA for time-series like LEAP(Yao et al., 2021), TDRL(Yao et al., 2022), NCTRL (Song et al., 2024). The formalization of block-wise identifiability on language-controlled components improves scalability and maintains theoretical guarantees, which requires a much smaller number of changes in auxiliary variables to identify the latent variables of interest than the previous work.

### A.6.3 APPLICABILITY

The applicability of WM3C is currently demonstrated in multi-task training and single-task adaptation settings in Meta-World. By pre-training the world model on a set of tasks, WM3C learns composable causal components while simultaneously training a multi-task policy. This composable representation not only makes policy learning more efficient but also enables quick adaptation to new tasks with shared components through minimal adjustments of components' dynamics. This makes WM3C particularly suitable for domains where tasks can be naturally decomposed into modular components, such as robotic manipulation or other scenarios with well-defined causal structures. Furthermore, while the current framework primarily uses language as the compositional modality, WM3C is expected to generalize to other modalities, such as visual or auditory signals. For example, auditory signals could be decomposed into frequency domains or audio patterns, allowing the framework to leverage the stability and generality of the composition system to handle multi-modal tasks effectively.

Current formulation of WM3C has limitations when applied to environments with highly overlapping causal components or complex language instructions (e.g., long or ambiguous sentences), where disentangling and identifying independent components becomes more challenging. However, these challenges are not insurmountable. One potential solution is to leverage large language models (LLMs) or human annotations to first convert long and intricate language descriptions into clearer, more structured formats. This preprocessing step simplifies the identification of compositional components by aligning the input with the framework's assumptions. Furthermore, causal components overlapping and complex structure learning (e.g. hierarchy) have been explored in the field of causal representation learning (Liu et al., 2023; Kong et al., 2023a; Morioka & Hyvärinen, 2023). By drawing on these existing studies, the WM3C framework can be extended to effectively

handle such scenarios. Jointly with the extension to offline learning, these enhancements would enable WM3C to be scale-up and learn a more complex and generalized causal component system.

## A.7 EXTENDED RESULTS

### A.7.1 SINGLE TASK PERFORMANCE IN META-WORLD

We present the success rate curves of all 18 training tasks, comparing WM3C, DreamerV3, and visual-based multi-task SAC. Results show that WM3C outperforms DreamerV3 in most tasks, even with CNN.

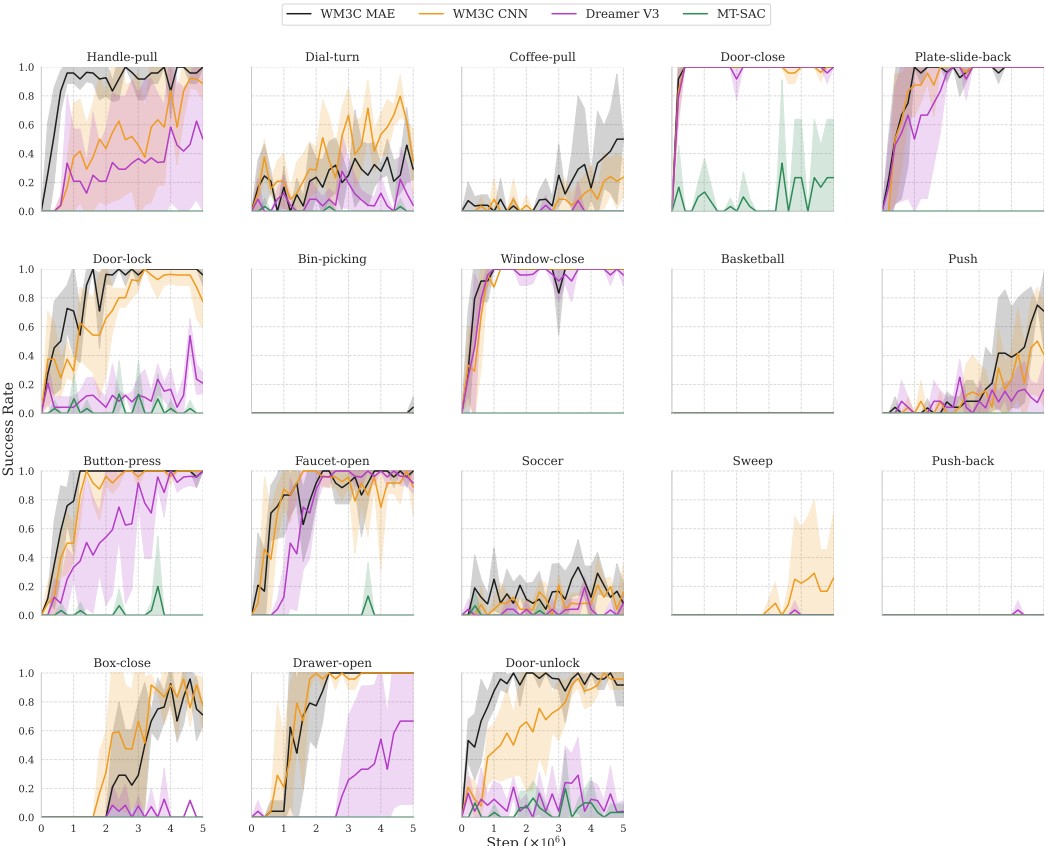

Figure 7: Training success rates on Meta-world: success rates curve of all 18 tasks

### A.7.2 ABLATION STUDY

To further investigate the effects of integrated modules, we conduct an ablation study on two scales of training in Meta-world, a 5 tasks training in 1M steps and a 18 tasks training in 2M steps. The 5 tasks are *push*, *faucet-open*, *handle-pull*, *door-lock* and *drawer-open*, a subset of complete 18 tasks. The 18 tasks are the same as those we use for training. We compare WM3C with MAE and CNN, and remove the mutual information constraints and decoding masks. The difference between the WM3C without masks and mutual information constraints and DreamerV3 is still present. WM3C features a language-conditioned, factorized dynamics module and learnable task embedding, which sets it apart from DreamerV3.

We see that all modules contribute to certain parts of the model's robust performance (see Figure 10). Using MAE as the visual module consistently improves the representation quality in both small and large-scale training regimes. The incorporated mutual information and sparsity constraints significantly improve sample efficiency consistently in both scales. Interestingly, we find that when

the task number increases, the benefits brought by the mutual information and sparsity constraints are larger.

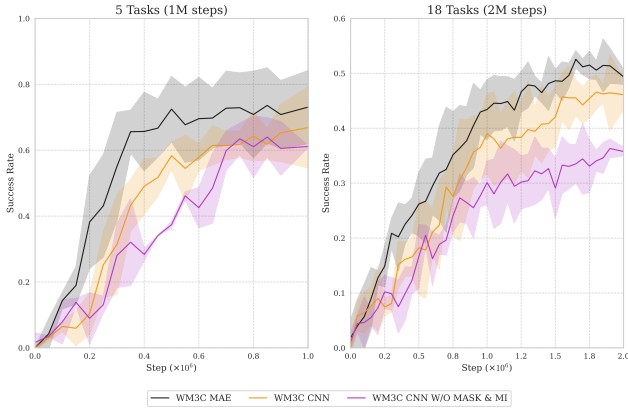

Figure 8: Ablation study on two scales of training data: 5 tasks in 1M steps training and 18 tasks in 2M steps training.

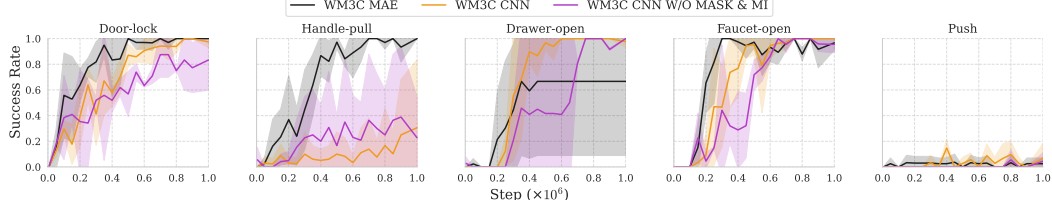

Figure 9: Individual tasks plot in 5 tasks and 1M steps training.

### A.7.3 INTERVENTION EFFECT ON THE LANGUAGE-CONTROLLED COMPONENTS

Here we present more examples of the effect of intervention on the language-controlled components at different tasks.

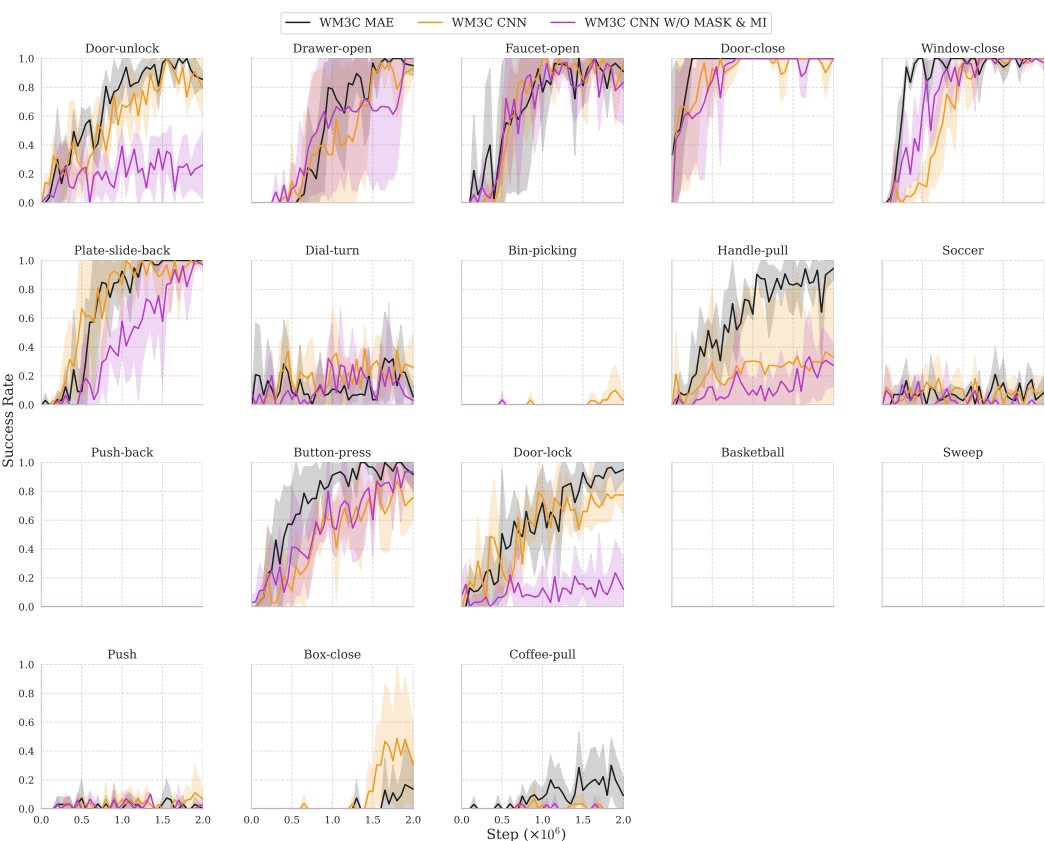

Figure 10: Individual tasks plot in 18 tasks and 2M steps training.

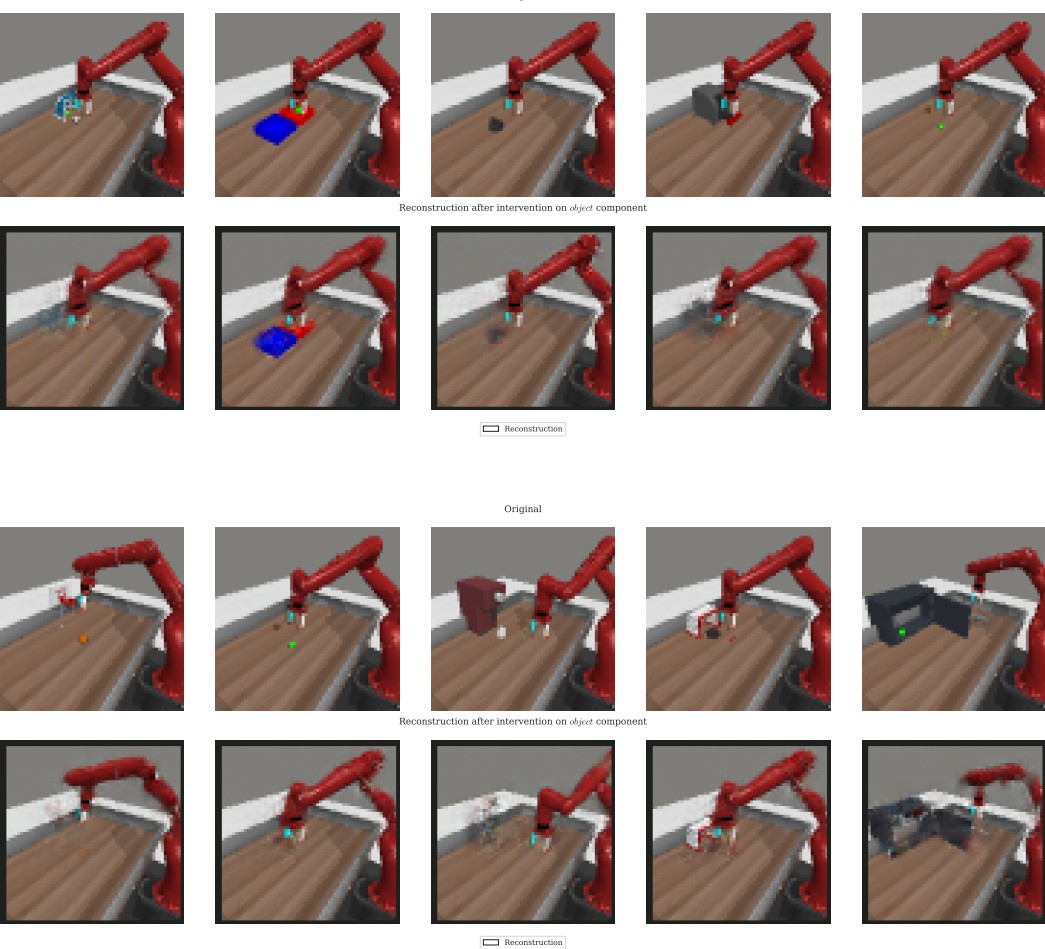

Figure 11: Intervention effect on the *object* component. The first row is the image observation WM3C receives, and the second row is the observation reconstruction from the latent whose object component has been intervened.

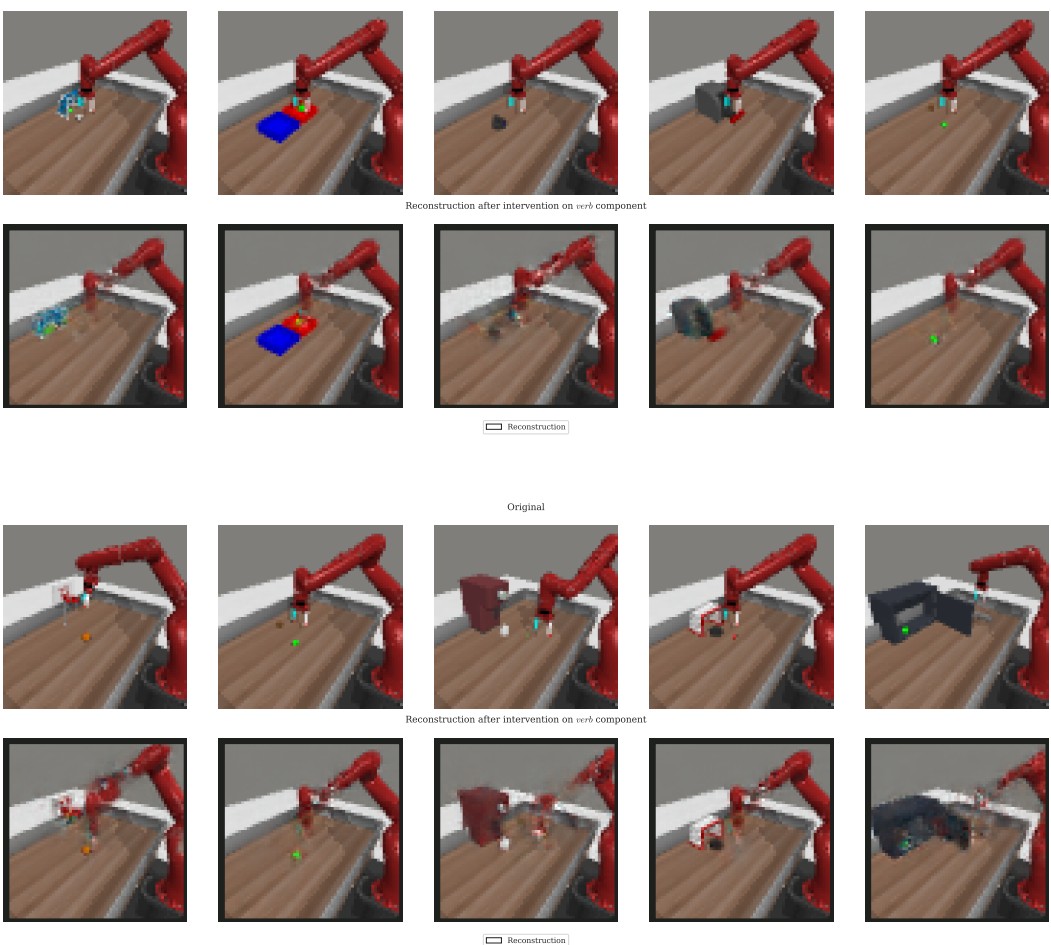

Figure 12: Intervention effect on the *verb* component. The first row is the image observation WM3C receives, and the second row is the observation reconstruction from the latent whose verb component has been intervened.

