# OpenReview forum: "Modeling Unseen Environments with Language-guided Composable Causal Components in Reinforcement Learning"
_ICLR.cc/2025/Conference — ICLR 2025 Poster_

### Official Review · Reviewer_mPjx · 2024-10-23

**Soundness:** 3
**Presentation:** 2
**Contribution:** 3
**Rating:** 6
**Confidence:** 3

**Summary:**

This paper introduces the World Modeling with Compositional Causal Components (WM3C) framework for reinforcement learning (RL). The framework aims to address the problem of generalization to unseen environments by leveraging compositional causal components by incorporating language-guided compositional causal reasoning inspired by human reasoning. Using language as a compositional modality, the framework decomposes latent space into semantically meaningful modular components with causal dynamics. Unlike prior methods focusing on invariant representation learning or meta-learning, WM3C aims to learn compositional components and utilize them to generalize effectively to new environments. The paper provides theoretical guarantees for the identifiability of these components under certain assumptions and implements the concept using a masked autoencoder (MAE) with mutual information constraints and adaptive sparsity regularization. WM3C's effectiveness is demonstrated through experiments on (synthetic) numerical simulations and (real-world) robotic manipulation tasks on the MetaWorld environment, where it outperforms existing approaches on generalization/adaptation to unseen tasks.

**Strengths:**

- **Originality**: WM3C's approach of integrating causal reasoning with compositional learning in RL is interesting. The use of language-guided decomposition adds a unique dimension to generalization in RL.

- **Quality**: Theoretical grounding is generally robust, with guarantees on the identifiability of compositional components. While the presented experimental results support the overarching claim, the strength and suitability of the experiments have room for improvements.

- **Clarity**: The clarity of the technical writing could be improved. Discussed further in the ‘Presentation’ section of the review.

- **Significance**: Generalization in RL is a well-known challenge. The paper tackles this critical problem in RL—generalization to unseen environments—and presents a solution with potential for significant impact in real-world applications.

**Weaknesses:**

**Discussion**
The crux of the work's novelty lies in the mapping between orthographic components (as we understand it: textual, visual, etc.) and latent space blocks, thereby allowing generalization to unseen tasks through the composition of these learned invariant primitive blocks. Such compositional generalization is a well-researched (and active) area, especially in the NLP, VQA, and EAI domains. For example, numerous papers using the CLEVR images and captions dataset, along with mixing the attributes in the VQA domain, explore such generalization using text and image modalities. One key enabler of these models was applying masked object encoders, thereby infusing the models with knowledge of what the objects are.
In this work, the effect of using MAE (vs. CNN) is not entirely clear or conclusive (from Fig. 5). However, the point remains: for the visual modality, the object and latent space mapping is fairly well-established and by no means novel, especially if object masking is in play.
The mapping between textual semantics — especially explicit mapping of complex verbs or actions with latent space blocks — is, however, quite promising and, to my knowledge, novel for this specific domain. However, the experiments conducted to show the efficacy use very simple instantiations of the language modality. There are a fairly large number of environments in the embodied AI space (like AI2Thor's ALFRED, Habitat, etc.) that have more realistic textual instructions for task completion scenarios—which, in my opinion (IMO), would be much more convincing to demonstrate the efficacy of this approach, the effectiveness of textual component identifiability, and the overall claim of the effectiveness of using causal RL.



* **Limited Evaluation** The presented experiments are weak. Experiment 1 using synthetic data (3 tokens, 3 control groups) should not be a major experiment with supporting figures in the main paper, but rather should have been a preliminary ‘smoke-test’ \- maybe getting a subsection of an appendix section at best. I have major concerns about the baselines of Experiment 2 conducted on MetaWorld’s robotic manipulation tasks.  The only baseline comparison is against DreamerV3 (2 variants).  I am wondering why any of the available, known, easily replicable RL models (e.g. MT-\[SAC|PPO\]) were not included as baselines?

* **Evaluation environments**: Given the claim of mapping ‘textual components’, the choice of environments and models could be more commensurate. Specifically, the proposed approach should be evaluated on environments that allow richer text+vision modality and agent interaction. There are a slew of such multi-modal environments across subdomains, like in the embodied AI domain: RoboThor (the ALFRED task), Habitat or, simpler HomeGrid. Given the richer, realistic text instructions on these domains, the claim of unique identification of causal textual components would be far better convincing.

* **Scalability**: The scalability of WM3C to environments with more complex dynamics and a higher number of compositional components has not been fully explored. Future work should test the framework in larger-scale environments.

* **Limited Analysis of Failure Cases**: It would be helpful to include a discussion on the limitations of WM3C, particularly regarding situations where the causal components may overlap or cannot be distinctly identified. Understanding where and why the model fails is crucial for improving the framework.

**Questions:**

**Questions**:

1. Can the authors elaborate on the train/test methodology of the baseline model (Dreamer V3) on MetaWorld? Were textual task descriptions included in the encodings? Did the authors use any existing implementation of Dreamer V3 on MetaWorld, or is it novel for this paper?

2. Why didn’t the authors include performance comparisons with generic RL models like Multitask-[SAC|PPO], which are provided out-of-the-box with the MetaWorld environment?

3. Could the authors provide more intuition or real-world examples where the faithfulness assumption might fail? How sensitive is the performance of WM3C to violations of this assumption?

4. What are the key limitations of WM3C in environments with highly overlapping causal components (more likely in richer/complex language instructions)? How might the model be adjusted to handle such scenarios?

5. How sensitive is the framework to the choice of language components? Would other compositional modalities (e.g., visual <-> auditory signals) work as effectively?

---

> ### Author Response · Authors · 2024-11-26
> **Response to Reviewer mPjx**
>
> Thank you for your insightful feedback. We have provided our responses to the weaknesses and questions below:
>
> > Numerous papers using the CLEVR images and captions dataset, along with mixing the attributes in the VQA domain...
>
> While compositional generalization has indeed been extensively studied in NLP, VQA, and EAI, our work **introduces two key contributions that set it apart**:
>
> * we provide theoretical guarantees and requirements for identifying language-controlled components from a causal perspective, which, to the best of our knowledge, is a novel contribution, and apply these principles to achieve compositional generalization in reinforcement learning.
>
> * regarding our masking approach, we employ standard MAE with random masking purely as an auxiliary technique to enhance semantic representation learning. Unlike previous work, we do not utilize object-specific or ground truth masks, but instead, we need to learn those mappings implicitly. Our hypothesis is that extracting language-corresponding information requires higher-level semantic understanding rather than low-level details, which MAE naturally facilitates. Importantly, our experiments with CNN-based WM3C demonstrate that the framework's effectiveness is not dependent on the choice of visual module.
>
> > Limited Evaluation: The presented experiments are weak...
>
> We appreciate the reviewer's thorough critique of our experimental evaluation. Let us address both points:
>
> Regarding the synthetic data experiment: this experiment serves a crucial purpose in validating our identification theory under controlled conditions where our assumptions can be verifiably fulfilled. Unlike the Meta-world environment, where ground truth latent states are inaccessible, this synthetic setup allows us to definitively verify our theoretical framework. This validation motivated our decision to include it in the main paper.
>
> Concerning the Meta-world baselines: We have expanded our comparison in the revised manuscript to include visual-based Multi-task SAC. We should note that many established Meta-world baselines, including standard MT-SAC, operate on state inputs rather than image observations, making direct comparisons inappropriate for our visual setting. We acknowledge that including additional vision-based baselines would strengthen our evaluation.
>
> > Evaluation environments: Given the claim of mapping ‘textual components’...
>
> We sincerely appreciate this valuable suggestion regarding richer multi-modal environments. We agree that environments like RoboThor (ALFRED), Habitat, and HomeGrid would provide more compelling demonstrations of our approach, particularly given their realistic text instructions and complex interactions.
>
> From a theoretical perspective, our current work demonstrates that unique identification is achievable for both textual components and discrete tokens under our stated assumptions, as validated through our simulation experiments. However, we acknowledge that testing in more sophisticated environments would strengthen our practical claims.
>
> We are actively working on extending WM3C to handle more complex structures and offline learning scenarios, with the goal of applying it to advanced embodied AI domains like ALFRED and Habitat. This represents an important direction for future work.
>
> > Can the authors elaborate on the train/test methodology...
>
> We have added detailed documentation of our DreamerV3 train/test methodology for Meta-world in the Appendix. Our implementation utilizes the official JAX codebase of DreamerV3, following the training protocols established in [1].
>
> [1] Ma, Haoyu, et al. "HarmonyDream: Task Harmonization Inside World Models." arXiv preprint arXiv:2310.00344 (2023).
>
> > Why didn’t the authors include performance comparisons with generic RL models...
>
> While Meta-world provides out-of-the-box implementations of MT-SAC and MT-PPO, these baseline models operate on state-based inputs rather than image observations. Since our method processes visual inputs, direct comparisons with these state-based implementations would not provide a fair evaluation.
>
> In our revised experiments, we have included visual-based Multi-task SAC as an additional baseline to provide a more comprehensive comparison within the vision-based setting.

---

> ### Author Response · Authors · 2024-11-26
> **Continual Response to Reviewer mPjx**
>
> > Could the authors provide more intuition or real-world examples where the faithfulness assumption might fail? How sensitive is the performance of WM3C to violations of this assumption?
>
> Thank you for this insightful question regarding faithfulness assumptions. Let us illustrate with a concrete example from robot manipulation:
>
> In multi-agent or multi-robot systems, the faithfulness assumption can be violated when forces or torques from different agents perfectly cancel each other out. This creates an apparent independence in observations that masks the true underlying causal relationships.
>
> While the faithfulness assumption is commonly used in causal studies, it's worth noting that under certain parameterizations, strict violations of faithfulness are mathematically rare [2][3].
>
> We acknowledge that a systematic study of how faithfulness violations affect causal RL performance would be valuable future work. This could provide important insights into the robustness of our approach under real-world conditions.
>
> [2] Zhang, Jiji, and Peter L. Spirtes. "Strong faithfulness and uniform consistency in causal inference." arXiv preprint arXiv:1212.2506 (2012).
>
> [3] Meek, Christopher. "Strong completeness and faithfulness in Bayesian networks." arXiv preprint arXiv:1302.4973 (2013).
>
> > What are the key limitations of WM3C in environments with highly overlapping causal components (more likely in richer/complex language instructions)? How might the model be adjusted to handle such scenarios?
>
> Thank you for this important question about handling complex, overlapping causal components. We identify two main limitations in our current approach:
>
> * Processing richer language instructions presents greater challenges than the relatively structured commands in Meta-world. However, we believe advances in large language models could help formalize and structure complex language components, potentially through hierarchical decomposition.
>
> * While our current formulation assumes clear component separation, it can be extended to handle overlapping connections and more complex latent structures by incorporating techniques from established causal representation learning work [4][5][6].
>
> We have added a detailed discussion of these limitations and potential extensions in Appendix 6.3.
>
> [4] Kong, Lingjing, et al. "Identification of nonlinear latent hierarchical models." Advances in Neural Information Processing Systems 36 (2023): 2010-2032.
>
> [5] Liu, Yuren, et al. "Learning world models with identifiable factorization." Advances in Neural Information Processing Systems 36 (2023): 31831-31864.
>
> [6] Kong, Lingjing, et al. "Learning Discrete Concepts in Latent Hierarchical Models." arXiv preprint arXiv:2406.00519 (2024).
>
> > How sensitive is the framework to the choice of language components? Would other compositional modalities (e.g., visual <-> auditory signals) work as effectively?
>
> Regarding sensitivity to language components, our framework is designed to work with any well-structured compositional system that maintains consistent semantic relationships. The key requirement is that the components should have clear, distinguishable roles in affecting the environment dynamics.
>
> As for other compositional modalities, this is an intriguing direction we are actively exploring. We hypothesize that any modality with stable compositional properties could be effectively integrated into our framework. For instance, auditory signals could be decomposed into distinct frequency domains or audio patterns. The effectiveness would likely depend on the stability and consistency of the compositional system - the more structured and reliable the decomposition, the more closely it should match the performance we see with language components.
>
> We are currently investigating the integration of different modalities to better understand the framework's flexibility and generalization capabilities across various compositional systems.

---

> ### Comment · Reviewer_mPjx · 2024-11-26
> **Reply to Author Responses**
>
> Dear Authors,
>
> I've reviewed the added experiments with `Multi-task SAC` (section 3.2 onwards), and appendix (A.5.2, specifically lines [1132-1165] that clarifies my question regarding implementation. (*My $0.02: In the future, kindly include the line numbers to addendum/specific changes for reducing cognitive load*).
>
> Thank you for your detailed responses. They included important clarifications that have significantly changed my outlook on your work. These include (please correct me if otherwise):
>
> - Reaffirmation that the MAE is learnt using random masking.
> - Use of visual inputs only sans state knowledge.
>
> While I still think the choice of environment for text <-> visual mapping (for the proposed work, hypotheses and goals) could be better, but this is nonetheless a promising step. I hope the authors will follow-through (as agreed as a future work direction) on real-word EAI domains, where it could be quite valuable.
>
> Based on your (excellent) responses, I've decided to raise my overall score for this work significantly.
>
> Best of luck!

---

> > ### Author Response · Authors · 2024-11-26
> >
> > Dear Reviewer mPjx,
> >
> > Thank you for your thorough assessment of our revised manuscript!
> >
> > We're pleased that our responses helped illuminate the key aspects of our approach, particularly regarding random masking in MAE learning and our use of visual inputs. Your suggestion to include line numbers for revisions is valuable feedback we'll implement in future submissions.
> >
> > Your point about the environment choice for more complex and richer text-visual mapping is well-taken. As noted, we are committed to extending this work to more real-world EAI domains, where we believe our approach will demonstrate substantial practical value.
> >
> > Thank you again for raising your rating and please let us know if you have additional concerns!
> >
> > Best,
> >
> > The Authors

---

### Official Review · Reviewer_Ytw5 · 2024-10-25

**Soundness:** 3
**Presentation:** 2
**Contribution:** 3
**Rating:** 6
**Confidence:** 3

**Summary:**

This paper proposes a method to identify the underlying composable causal components in reinforcement learning (RL) environments to facilitate policy generalization. The main idea is to leverage language descriptions of the task to identify the underlying latents that govern the data generation process. The paper implements this theoretical result based on the model-based RL framework Dreamer V3, with additional constraints to foster sparsity and modularity of the learned components. The proposed method is empirically verified on synthetic data and robot manipulation tasks in Meta-World.

**Strengths:**

- The identification result in Theorem 1 is interesting and implies an advantage of language-driven RL agents compared to non-language counterparts.

- Empirical results are generally good. I also like the intervention results in Figure 6.

**Weaknesses:**

I think the biggest weakness of the paper in its current form is the discrepancy between the motivation, theoretical results, and the algorithm.
- The authors motivate their algorithm by the need to identify composition causal components in the underlying data generation model, which possesses some ideal traits like modularity and sparsity, as introduced in the causal learning literature. However, if our aim is to identify "causal" components, then we need to recover not only those components but also their causal structure (as also noted by the authors in Lines 74-75), or the word "causal" itself would become meaningless. Yet, the identification result presented in Theorem 1 gives no guarantee of recovering such a causal structure.

- On the other hand, the algorithm indeed uses modularity and sparsity constraints. However, such constraints are not reflected in the identification results, making the theory and the algorithm not really consistent. It feels to me that those constraints are more heuristics inspired by the causal learning literature, rather than motivated by Theorem 1.

I think to rectify the potential understanding, the authors should explicitly make it clear that the theoretical results do not guarantee "causal" identification.

Also, I think the authors should make the relationship between their theoretical results and prior results in representation identifiability more clear.

- The authors have claimed that prior methods for identifying latent variables use "different priors like functional form prior, independent noise conditions and auxiliary variables" (Line 176), which limits their scalability. Yet, the "language components" used by the paper can also be modeled as "auxiliary variables", which indeed play a crucial role in the identifiability result.

- Technically, the high-level idea of proving identifiability is similar to the results in [1, 2], which rely heavily on the conditional independence between latent variables and observable variables. The authors could include a discussion of those results.

[1] Nonlinear ICA Using Auxiliary Variables and Generalized Contrastive Learning.

[2] Variational Autoencoders and Nonlinear ICA: A Unifying Framework.

**Questions:**

- What is the relation between the identifiability result in Theorem 1 and prior results in non-linear ICA?

- What is the effect of each constraint in the algorithm? It seems that no ablation on them is presented in the experiments section.

Minor:

- Typo: Line 252 -> bfrom.

---

> ### Author Response · Authors · 2024-11-26
> **Response to Reviewer Ytw5**
>
> We appreciate your valuable feedback, and we arrange the response to your concerns and questions as follows:
>
> > The authors motivate their algorithm by the need to identify composition causal components…
>
> Thank you for your thoughtful comments. We’d like to clarify that once latent variables are identified using Theorem 1, the identifiability of the causal structure becomes straightforward and can be directly achieved using existing causal discovery methods. This is why we focus on emphasizing the identifiability of the latent variables, as in [1][2][3]. Following your suggestion, we have added a sentence discussing the identifiability of causal structures in the manuscript. Moreover, in the estimation, causal structures are learned at the same time by treating them as free parameters.
>
> [1] Yao, Weiran, et al. "Learning temporally causal latent processes from general temporal data." arXiv preprint arXiv:2110.05428 (2021).
>
> [2] Song, Xiangchen, et al. "Temporally disentangled representation learning under unknown nonstationarity." Advances in Neural Information Processing Systems 36 (2024).
>
> [3] Yao, Weiran, Guangyi Chen, and Kun Zhang. "Temporally disentangled representation learning." Advances in Neural Information Processing Systems 35 (2022): 26492-26503.
>
> > On the other hand, the algorithm indeed uses modularity and sparsity constraints…
>
> Thanks for the comments. Let us clarify it below. Theorem 1 is about the theoretical identifiability when there are **sufficient samples**. Note that we do not require the modularity and sparsity assumption to achieve theoretical identifiability, but instead, they are properties of the causal system [4][5]. During practical estimation, we apply the L1 norm to the binary causal masks to suppress the small entries closer to zero when dealing with **finite samples**, which is widely used in dealing with finite samples. Regarding the mutual information constraint, we build a connection between causal structure and independence, so that we can make use of the statistical constraints to learn different components.
>
> [4] Perry, Ronan, Julius Von Kügelgen, and Bernhard Schölkopf. "Causal discovery in heterogeneous environments under the sparse mechanism shift hypothesis." Advances in Neural Information Processing Systems 35 (2022): 10904-10917.
>
> [5] Schölkopf, Bernhard, et al. "Toward causal representation learning." Proceedings of the IEEE 109.5 (2021): 612-634.
>
> > What is the relation between the identifiability result in Theorem 1 and prior results in non-linear ICA?
>
> Thank you for this important question about the relationship to non-linear ICA. While our work builds upon valuable insights from them, such as [1][2][3][6][7], there are two key distinctions:
>
> * Our identification proof extends beyond traditional approaches by addressing multiple co-existing language components within a POMDP framework, where each component independently controls its corresponding latent variables. This differs from prior results which typically consider simpler graphs with a single auxiliary variable connected to all latent variables. Additionally, our approach does not require the parametric assumptions (such as conditional exponential) found in [6][7].
>
> * Our focus on block-wise identification of language-controlled components offers practical advantages, requiring significantly fewer language component values compared to traditional non-linear ICA approaches [1][3][7].
>
> We have added a comprehensive comparison with previous non-linear ICA work in Appendix 6.2.
>
> [6] Nonlinear ICA Using Auxiliary Variables and Generalized Contrastive Learning.
>
> [7] Variational Autoencoders and Nonlinear ICA: A Unifying Framework.
>
> > What is the effect of each constraint in the algorithm? It seems that no ablation on them is presented in the experiments section.
>
> We have added the comparison between using MAE and CNN as visual modules, the ablation of the mutual information constraints and masks in two different scales of training in Appendix 7.2. It shows that each module contributes to some aspects of the model learning.

---

> > ### Comment · Reviewer_Ytw5 · 2024-11-26
> > **Reviewer's response**
> >
> > Thank you for your response, which has addressed most of my initial concerns. In response, I have raised my rating from 5 to 6. In the future version of the paper, I suggest incorporating the clarifications of Theorem 1 on causal discovery and the comparison with prior non-linear ICA results into the main text for better readability.

---

> ### Author Response · Authors · 2024-11-26
>
> Dear Reviewer Ytw5,
>
> Thank you for raising your rating and your feedback!
>
> >  I suggest incorporating the clarifications...
>
> We will adjust the main text to incorporate the clarifications of Theorem 1 on causal discovery and the comparison with prior non-linear ICA results.
>
> Please let us know if you have additional concerns!
>
> Best,
>
> The Authors

---

### Official Review · Reviewer_ByBw · 2024-10-27

**Soundness:** 3
**Presentation:** 3
**Contribution:** 3
**Rating:** 6
**Confidence:** 3

**Summary:**

The paper presents World Modeling with Compositional Causal Components (WM3C), a novel approach in reinforcement learning (RL) aimed at improving generalisation in novel environments. Inspired by human reasoning, WM3C identifies and utilizes causal, compositional elements, leveraging language as a guiding modality. The approach builds on the causal structure of RL tasks by combining theoretical identifiability guarantees with a masked autoencoder architecture and adaptive sparsity. WM3C outperforms existing methods in identifying causal components and supports efficient policy learning and generalization, as shown through synthetic data simulations and robotic manipulation tasks in the Meta-World environment. Although promising, the results’ scalability in more diverse real-world scenarios remains unclear.

**Strengths:**

1. An exciting and important problem is being tackled.
2. Comprehensive conceptual and theoretical analysis.

**Weaknesses:**

My main concern is a relatively lean experimental section; see the questions and points below.

1. As the proposed method is quite complicated, it is unclear if the effects can be due to the described mechanisms.
2. Only one real-world environment is tested (I am aware that it might be partially due to a lack of proper benchmarks).
3. Only one algorithm is tested.
4. The transfer and generalization results are relatively weak (at the same time, I am not quite sure if the current scale of experiments is large enough to reveal strong effects.)

**Questions:**

1. How is the presented setup related to hierarchical RL?
2. How is it possible that WM3C can acquire the latent language structure with only 20 tasks? It feels very nice but somewhat implausible.
3. Adaptation results in Fig 5 feel somewhat poor. Could you comment on that?
	1. I'm wondering if pure Dreamer is a proper baseline. E.g., recent [1] shows that transfer is better if forgetting mitigation is applied
4. Could you describe the applicability of the proposed approach? Is it a multi-task setting like Meta-World or perhaps pertaining?

[1] Fine-tuning Reinforcement Learning Models is Secretly a Forgetting Mitigation Problem, Wolczyk et al.

---

> ### Author Response · Authors · 2024-11-26
> **Response to Reviewer ByBw**
>
> Thank you for your valuable feedback, our response to your concerns and questions are arranged as follows:
>
> > As the proposed method is quite complicated, it is unclear if the effects can be due to the described mechanisms.
>
> We appreciate the reviewer's concern regarding our method's complexity and the attribution of its effects. Let us clarify them below:
> * To validate our mechanisms, we have conducted simulation experiments demonstrating their role in both i.i.d latent identification and o.o.d prediction. Additionally, we have added an ablation study in Appendix 7.2 comparing integrated modules with vanilla WM3C at different training scales.
> * More importantly, while our method includes multiple components, the modifications are straightforward, and our results show they provide clear benefits to improving learning and adaptation.
>
> > Only one real-world environment is tested (I am aware that it might be partially due to a lack of proper benchmarks).
>
> We also resonate that there aren’t many proper benchmarks supporting a diverse and large number of compositional tasks.
>
> > Only one algorithm is tested.
>
> In our Meta-world experiments, following your suggestion, we have included visual-based Multi-task SAC as a model-free baseline. Additionally, we would like to mention that we selected DreamerV3 as our primary comparison because it represents the state-of-the-art, offering strong sample efficiency and generalization across various scenarios. We believe DreamerV3 is a sufficiently robust and representative comparison point for evaluating the effectiveness of our approach.
>
> > How is the presented setup related to hierarchical RL?
>
> Thank you for this insightful question about the relationship to hierarchical RL. While both approaches involve decomposition, they differ fundamentally in their structure and objectives. Hierarchical RL decomposes tasks temporally, with high-level policies orchestrating low-level policies through subtask selection. In contrast, our approach focuses on the spatial-temporal decomposition of task dynamics into causally independent components (such as "verb" and "object"), emphasizing modular recombination rather than temporal hierarchy.
>
> For a more detailed comparison of these approaches, we have added a comprehensive discussion in Appendix 6.1.
>
> > How is it possible that WM3C can acquire the latent language structure with only 20 tasks? It feels very nice but somewhat implausible.
>
> The surprisingly effective performance with only 20 tasks can be explained by examining the underlying structure of our MDPs:
>
> The dimensionality of each true latent component is relatively compact. For instance, the robot state comprises just 9 dimensions (7 joint positions, 1 gripper position, and 1 gripper velocity), while object states are characterized by 6 dimensions (3D position and 3D orientation). Based on our identifiability theory for language-controlled components (Theorem 1), we only need 10 and 7 distinct changes in the corresponding language component values for identification, respectively.
>
> Furthermore, our implementation leverages KL divergence and mutual information constraints, which actively promote the separation of latent spaces between distinct components, enhancing the efficiency of the identification process.
>
> > The transfer and generalization results are relatively weak
>
> > Adaptation results in Fig 5 feel somewhat poor.
>
> Thanks for the thoughtful comments. We have modified the setup of the training and adaptation in Meta-world so that we can have a more meaningful comparison. Concretely, we increase the number of test tasks and evaluate them in a longer training time instead of limiting the adaptation steps used. In the new adaptation experiments, we see that with partial fine-tuning, WM3C outperforms full-parameter-tuned DreamerV3 and MT-SAC noticeably (see Figure 5).

---

> ### Author Response · Authors · 2024-11-26
> **Continual Response to Reviewer ByBw**
>
> > I'm wondering if pure Dreamer is a proper baseline. E.g., recent [1] shows that transfer is better if forgetting mitigation is applied
>
> Thank you for bringing attention to [1]. Indeed, forgetting mitigation techniques could enhance both DreamerV3 and WM3C's policy modules during fine-tuning. However, we would like to clarify two key distinctions in our research focus:
>
> * Our primary contribution lies in developing a modular environment model to facilitate policy adaptation in the POMDP, whereas [1] primarily addresses catastrophic forgetting in pretrained model-free policies.
>
> * We selected DreamerV3 as our model-based baseline due to its well-documented stability and learning efficiency across diverse environments. Our revised adaptation experiments demonstrate that even without forgetting mitigation, DreamerV3 achieves competitive performance.
>
> [1] Fine-tuning Reinforcement Learning Models is Secretly a Forgetting Mitigation Problem, Wolczyk et al.
>
> > Could you describe the applicability of the proposed approach? Is it a multi-task setting like Meta-World or perhaps pertaining?
>
> Thank you for pointing this out, we have added it to a section of Appendix 6.3. In the current formulation, it is a multi-task pretraining that trains the world model on a bundle of tasks and learns the components. At the same time, the policy is trained as a multi-task policy model. Further, the pre-trained models can be adapted to a new task that has known components with only a few samples and changes in the model. In the future, we would extend it to offline learning, where we could leverage video data to learn a more generalized and comprehensive component system.

---

> ### Comment · Area_Chair_xC51 · 2024-11-27
> **Rebuttal Response**
>
> Dear Reviewer,
> Do you mind letting the authors know if their rebuttal has addressed your concerns and questions? Thanks!
> -AC

---

> ### Author Response · Authors · 2024-11-27
>
> Dear Reviewer ByBw,
>
> We want to kindly follow up regarding our rebuttal. We truly appreciate the time and effort you’ve already dedicated to reviewing our work and providing your feedback.
>
> To address your main concern on the "relatively lean experimental section":
>
> * We included Multi-task SAC as an additional baseline to the Meta-world experiments (Figure 4 and Figure 5) and added an ablation study about mutual information constraints and masks (Appendix 7.2). We also clarify that the i.i.d identification and o.o.d prediction can validate our described mechanisms
> * We modified the adaptation setting to show the superior generalization ability of WM3C against DreamerV3 and Multi-task SAC (Figure 5, where WM3C CNN surpasses DreamerV3 noticeably on almost all 9 test tasks).
> * We added discussion sections of both hierarchical RL (Appendix 6.1) and applicability (Appendix 6.3).
>
> You could review more detailed responses about your concerns and questions in the previous comments above, could you please let us know if our responses above can address your concerns and questions? Thank you!
>
> Best,
>
> The Authors

---

> > ### Comment · Reviewer_ByBw · 2024-12-02
> > **Thank you**
> >
> > I acknowledge reading the rebuttal. Thank you for your work. I am still in favour of this paper being accepted.

---

> > > ### Author Response · Authors · 2024-12-02
> > >
> > > Dear Reviewer ByBw,
> > >
> > > Thank you for taking the time to review our rebuttal and for your support in favor of accepting our work.
> > >
> > > Given the additional experiments, analyses, and clarifications we provided to address your concerns, we kindly ask if you might consider raising your score to further reflect the improvements and contributions of our work.
> > >
> > > Your support and evaluation mean a great deal to us, and we sincerely thank you again for your valuable input and time.
> > >
> > > Best regards,
> > >
> > > The Authors

---

### Official Review · Reviewer_svqL · 2024-11-04

**Soundness:** 3
**Presentation:** 2
**Contribution:** 2
**Rating:** 6
**Confidence:** 3

**Summary:**

After reading the rebuttal, I agree that language information could be somewhat generally available and I appreciate the effort in fixing writing issues with the paper, so I decided to slightly increase my score.

This paper proposes a verb/object decomposed world model where the verb and object for a task are given beforehand.  There is improved sample complexity on metaworld, but the method seems like it’s much more complicated and using quite a bit of domain knowledge.  The analytical experiments are good.

notes from reading the paper:
  -RL generalization to new environments is difficult, especially if there are unseen dynamics.
  -World modeling with composition causal components.
  -Causal dynamics over composable elements.
  -Language as a compositional modality.
  -Masked autoencoder with MI constraint and sparsity to capture high-level semantic information and disentangle transition dynamics.
  -Experiments on real robots and numerical simulations, method is called WM3C.
  -DRL algorithms have a difficult time to adapt to changes in the environment.  Many ways of getting around this problem require domain-specific knowledge.
  -Composable causal components and dynamics can be uniquely identified (this seems surprising to me).
  -Language-controlled components are a subset of state dimensions that are directly controlled by individual language components.
  -[o_t, r_t] = g(s_t, \eps_t).  s_t is defined as set of composable components.  The c_{it} is defined as a distribution conditioned on the language l_t and the previous state and action.
  -Blockwise identifiability requires the mixing function to be invertible and smooth.  Also every state must have density.
  -Utilize model-based RL algorithm DreamerV3.
  -Additional sparsity constraint.
  -Only reward-relevant states are used for the policy model.

**Strengths:**

Good analytical experiments, some improvements in sample complexity on metaworld.

**Weaknesses:**

Method adds substantial complexity and also seems to require much more domain knowledge than the baseline.

**Questions:**

-Typo on first page, poorly is spelled “pooly”.
  -How much effort needs to go into figuring out the “controllable composable components”?  This might be less desirable than a system which works for any general MDP, by processing the observations.
   -Experiments show effective component identification over the course of training in a synthetic task.
  -The gains in sample complexity on meta-world still seem to be significant.
  -Figure 6 looks fairly unprofessional and is hard to make sense of.  Intervention the verb component makes the arm blurry, which is good to verify.  However, it doesn’t really clearly show a change what the action or the object are.  This may be related to the nature of pixel-space reconstructions.

---

> ### Author Response · Authors · 2024-11-26
> **Response to Reviewer svqL**
>
> Thank you for your constructive feedback. We have tried to address all your concerns as follows.
>
> > Method adds substantial complexity and also seems to require much more domain knowledge than the baseline.
>
> Thanks for the thoughtful comments. We would like to clarify that this domain knowledge we utilize, primarily in the form of language descriptions, is relatively straightforward to obtain and cost-effective. In fact, modern LLMs can assist in generating this knowledge. Our experimental results demonstrate that this modest investment in domain knowledge leads to substantial benefits in terms of sample efficiency during both training and adaptation phases of WM3C, especially when compared to baselines such as DreamerV3 and MT-SAC. We believe this trade-off between initial knowledge input and improved performance represents a valuable contribution to the field.
>
> > Typo on first page, poorly is spelled “pooly”.
>
> > Figure 6 looks fairly unprofessional and is hard to make sense of. Intervention the verb component makes the arm blurry, which is good to verify. However, it doesn’t really clearly show a change what the action or the object are. This may be related to the nature of pixel-space reconstructions.
>
> Following your valuable feedback, we have thoroughly addressed the typos throughout the manuscript. Additionally, we have enhanced the clarity of Figure 6 by incorporating detailed captions and visual guides. Specifically, in the revised version, we have added clear indicators to highlight the image regions that demonstrate the effects of interventions on different language-controlled components. These improvements should make the results more accessible and easier to interpret.
>
> > How much effort needs to go into figuring out the “controllable composable components”? This might be less desirable than a system which works for any general MDP, by processing the observations.
>
> A main intuition our identification results showed is that under some mild assumptions of the data generation process, as long as we have a sufficient number of distinct tasks (containing sufficient changes of all the language components), figuring out the controllable composable components is natural with some lightweight changes in the architecture and optimization of the world model. That being said, as we scale up the data, this is fulfilled gradually easily, and reasonably. In our newly added experiments, we see that, as a typical baseline working for any general MDP, Multi-task SAC is outperformed by WM3C (see Figure 4 and 5), which demonstrates the benefits of learning such modular and controllable components.

---

> ### Comment · Area_Chair_xC51 · 2024-11-27
> **Rebuttal Response**
>
> Dear Reviewer,
> Do you mind letting the authors know if their rebuttal has addressed your concerns and questions? Thanks!
> -AC

---

> ### Author Response · Authors · 2024-11-27
>
> Dear Reviewer svqL,
>
> Could you please let us know if our responses above can address your concerns and questions? Thank you!
>
> Best,
>
> The Authors

---

> ### Author Response · Authors · 2024-11-28
>
> Dear Reviewer svqL,
>
> We want to kindly follow up regarding our rebuttal. We truly appreciate the time and effort you’ve already dedicated to reviewing our work and providing your feedback.
>
> We’ve worked to address your concerns and questions as follows:
>
> * **Complexity and Domain Knowledge**: We clarified that the domain knowledge used in our approach, primarily language descriptions, is cost-effective and straightforward to obtain, with assistance from modern LLMs if needed. This modest investment results in substantial benefits for sample efficiency during both training and adaptation, as demonstrated in our experiments.
>
> * **Figure 6 and Presentation**: Based on your feedback, we enhanced Figure 6 by adding detailed captions and visual guides to highlight the effects of interventions on language-controlled components, making the results clearer and more interpretable.
>
> * **Effort in Identifying Controllable Components**: We provided additional intuition and experimental evidence showing that under mild assumptions, identifying controllable components becomes natural and scalable with sufficient task diversity. Furthermore, new experiments demonstrate WM3C outperforming general MDP baselines like Multi-task SAC (Figures 4 and 5), underscoring the benefits of learning modular and controllable components.
>
> * **Typographical Errors**: We addressed the typo and thoroughly reviewed the manuscript for similar issues.
>
> You could review more detailed responses about your concerns and questions in the previous comments above. We’d greatly appreciate it if you could let us know whether our revisions have addressed your concerns. Thank you!
>
> Best,
>
> The Authors

---

### Author Response · Authors · 2024-11-26
**Review Summary**

We sincerely thank all reviewers for their constructive feedback and insightful comments. We are encouraged by the recognition of the novelty and significance of WM3C's contribution to improving RL generalization via the unique integration of language-guided decomposition for causal compositional learning by Reviewer **ByBw**, and **mPjx**, the emphasization of the comprehensive conceptual and theoretical analysis by Reviewer **Ytw5** and **mPjx**, the acknowledgment of the empirical verification of WM3C, including strong sample efficiency and component identification by Reviewer **svqL** and **Ytw5**. The theoretical rigor, which Reviewer **ByBw** and **Ytw5** highlight, directly addresses Reviewer **mPjx**'s concerns about novelty by establishing formal guarantees for component identification that extend beyond existing work in NLP/VQA domains. While Reviewer **mPjx** suggested our experiments were "weak”, Reviewer **svQL** and **Ytw5** positively noted that our analytical experiments and empirical results are good. All reviewers agreed on the importance of addressing RL generalization and the potential impact of our contributions to this challenging and critical area of research.

Below is a summary of our responses:

* **Changes**: We  modified the adaptation setting to better demonstrate the superiority of WM3C regarding adaptation. We added multi-task SAC as a model-free baseline to the Meta-world experiments and added an ablation study of WM3C in the Appendix. We also added extended discussions and model/training/adaptation details in the Appendix.

* **To Reviewer svqL**: We clarified the trade-off between complexity and performance benefits, refined Figure 6 for clarity, and demonstrated how identifying composable components can emerge naturally under our theoretical framework.

* **To Reviewer ByBw**: We added multi-task SAC as a baseline, conducted additional ablation studies, and extended adaptation experiments to show stronger generalization results. We also clarified the relationship between WM3C and hierarchical RL in Appendix 6.1.

* **To Reviewer Ytw5**: We clarified that, after applying our identifiability results to the latent variables, existing causal discovery identifiability results could be directly leveraged to establish the identifiability of the structure among latent variables.   We also provided a detailed comparison with prior work in non-linear ICA and added ablation studies in Appendix 7.2.

* **To Reviewer mPjx**: We included new baselines and highlighted the importance of synthetic experiments for validating theoretical claims. We acknowledged the need for richer multi-modal environments and discussed potential extensions for overlapping causal components in Appendix 6.3.

Please review our detailed responses to each point raised. We have highlighted the revisions in different colors to correspond to specific reviewers' comments. We hope these revisions and clarifications effectively address your concerns. Thank you once again for your valuable time and insightful feedback.

---

### Meta-Review · Area_Chair_xC51 · 2024-12-18

**Metareview:**

**Summary** This paper studies compositional generalization in reinforcement learning.  The method, World Modeling with Compositional Causal Components (WM3C), focuses on identifying language-based latent variables which can be used to create a causal world model for the environment.  This structure enables policies which can  generalize to new tasks.

**Strengths** Reviewers highlighted the strong experiments which gave improved performance on metaworld and showed the ability of the model to learn useful and meaningful latents, e.g. by interventions on the latent variables.  At the same time, the paper also contains interesting theoretical analysis on the identifiability of language-controlled components, which underpins the method.

**Weaknesses** Some reviewers noted the complexity of the model as a drawback.  There were also questions related to the required domain knowledge and additional supervision.  During the rebuttal the authors made the argument that the language information is not a large bottleneck for the method and I and the reviewers generally agreed.  There were also questions about a gap between the theory and the actual algorithm which the authors did a good job filling during revision.  Lastly, some reviewers pointed out the experimental evaluation could be more comprehensive with more than one real-world experiment and more ablations and baselines.  The authors added a baseline and ablations. I agree this may be weaker side of the paper, but I think the evaluations are sufficient.

**Conclusion** The reviewers and AC are unanimous for acceptance.  The paper provides an interesting method to tackle an important problem and is backed by good theory and sufficient empirical evaluation.

**Additional Comments On Reviewer Discussion:**

svqL increased their score from 5 to 6 in response to improvements in writing and the authors’ argument that language descriptions could be found to train the model.   ByBw suggested the need for better attribution for the methods benefits the authors added an ablation to address this.  ByBw suggested the need for additional baselines and the authors added one.

---

### Decision · Program_Chairs · 2025-01-22

Accept (Poster)